# Ensemble representations reveal distinct neural coding of visual working memory

Byung-Il Oh [1], Yee-Joon Kim [2] & Min-Suk Kang [1,3]*

We characterized the population-level neural coding of ensemble representations in visual working memory from human electroencephalography. Ensemble representations provide a unique opportunity to investigate structured representations of working memory because the visual system encodes high-order summary statistics as well as noisy sensory inputs in a hierarchical manner. Here, we consistently observe stable coding of simple features as well as the ensemble mean in frontocentral electrodes, which even correlated with behavioral indices of the ensemble across individuals. In occipitoparietal electrodes, however, we find that remembered features are dynamically coded over time, whereas neural coding of the ensemble mean is absent in the old/new judgment task. In contrast, both dynamic and stable coding are found in the continuous estimation task. Our findings suggest that the prefrontal cortex holds behaviorally relevant abstract representations while visual representations in posterior and visual areas are modulated by the task demands.

[1] Department of Psychology, Sungkyunkwan University, 25-2 Sungkyunkwan-ro, Jongno-gu, Seoul 03063, South Korea. [2] Center for Cognition and Sociality, Institute for Basic Science (IBS), 55 Expo-ro, Yuseong-gu, Daejeon 34126, South Korea. [3] Center for Neuroscience Imaging Research, Institute for Basic Science (IBS), 2066 Seobu-ro, Jangan-gu, Suwon 16149, South Korea. *email: kangminsuk@skku.edu

Visual working memory is a capacity-limited, temporary buffer that maintains and manipulates visual information for a short period[1,2]. Growing evidence indicates that visual working memory is distributed over multiple brain regions[3–7]. In particular, the prefrontal cortex (PFC) and posterior parietal cortex (PPC) have been the focus of recent research efforts. One model proposes a posterior-to-frontal axis gradient such that behaviorally relevant information is gradually abstracted from sensory areas to the PFC[3,4,8,9]. This model is appealing considering the fact that other types of information are similarly organized; abstract information about an object is represented in the anterior region of the temporal cortex[10] and the hippocampus[11], and complex control of information occurs in the anterior region of the PFC[12]. However, other studies have also shown that the neuronal selectivity within the PFC and PPC change similarly according to the goal [13–17], such that they both hold stimulus-specific mnemonic representations of simple features[18] as well as abstract representations of category[19,20].

Here, we investigate how visual working memory representations are structured in distributed brain regions by ensemble, which provides a unique opportunity for studying the structured representation of working memory. Specifically, when we have to hold more than one item, we do not store them independently. Instead, our visual system builds a hierarchical structure of visual working memory by grouping similar items together through a series of cognitive operations[21,22], and utilizes the ensemble to better remember the items in visual working memory[23–26]. This means that while participants perform a single task, the visual system encodes high-order information such as ensemble mean and its variance, in addition to noisy sensory inputs[23–25]. This feature makes the ensemble unique for studying structured representations of visual working memory, and distinct from previous studies that used stimuli in which abstract and sensory information are nearly indistinguishable[3,18], or that adopted different tasks for multi-featured stimuli, requiring participants to select a particular dimension by which to identify different types of representations[19].

To capture the neural representations of ensemble in working memory, we develop an experimental procedure with two advancements. First, to utilize an inverted encoding model (IEM)[27,28] that can reconstruct single as well as multiple items in their feature space[29,30], we specially devise a set of stimuli. In one condition, participants are presented with 20 objects which are the same in their orientation (same orientation, SO). From a subset of these trials, we train an orientation decoder and then generalize it to the remaining trials to reconstruct mental representations of the SO condition. In the second condition, we present the participants with 20 objects whose orientations are varied around the predetermined mean orientation (varied orientation, VO). We then generalize the decoder of the SO condition to the trials of the VO condition. This generalization reconstructs the hierarchically encoded ensemble as well as item representations from the VO condition.

Second, we evaluate the spatiotemporal dynamics of visual working memory. Recent advances applying multivariate decoding algorithms in conjunction with high-temporal resolution electroencephalography (EEG) have enabled us to examine the distinctive temporal dynamics of neural representation, especially whether its coding is stable or dynamic[31]. Stable coding refers to a situation where a content-specific representation is coded by stable neural patterns over time, so that the pattern at one time can also reconstruct the same representation at another time. On the other hand, dynamic coding refers to a situation whereby dynamically changing neural patterns code content-specific representations which can only be reconstructed at the time point when the decoder is built. Theoretical roles of the temporal profile of working memory have been extensively discussed[7,15,32,33], and we question whether neural representations in different hierarchies rely on the same coding scheme. Further, to investigate the spatiotemporal dynamics, we apply this method to frontocentral and occipitoparietal electrodes separately and quantify the dynamics with a newly developed stable/dynamic index.

Our results show that a set of similar items is stored in a structured manner in different brain areas. We first confirm that we can reconstruct neural representations of simple features as well as ensemble statistics, and identify their temporal dynamics in visual working memory. We then investigate their spatiotemporal organization using two tasks with different demands. We find that the frontocentral areas stably coded simple features as well as mean orientations, and the neural representation of ensembles even correlated with behavior. In contrast, task demand modulates the neural coding of ensembles at the occipitoparietal areas, which does not correlate with behavior. Taken together, these results suggest that the frontocentral areas store behaviorally relevant abstract working memory representations, whereas the occipitoparietal areas store visual representations modulated by top-down task demands.

## Results

**Stimuli, tasks, and behavioral results.** We asked participants to remember 20 orientations, which were carefully designed to decode their neural correlates (Fig. 1a). Specifically, we first determined a mean orientation from the predefined set of orientations from −78.75°, −56.25°, −33.75°, −11.25°, 11.25°, 33.75°, 56.25°, 78.75°. We then determined a set of orientations for the same-orientation (SO) condition and for the varied-orientation (VO) condition. In the SO condition, all 20 orientations were the same as the mean orientation. In the VO condition, orientations were varied from the mean orientation by −30°, −10°, +10°, and +30° (Experiment 1) or −22.5°, −7.5°, +7.5°, and +22.5° (Experiment 2), with five repetitions of each orientation. Note that in the VO condition, the mean orientation was not presented. These two conditions were presented to the participants in a blocked manner in both experiments.

In Experiment 1, 20 participants performed an old/new judgment task. Specifically, the participants were presented with the 20 oriented bars for 100 ms, followed by the blank retention interval for 1500 ms in both the SO and VO conditions (Fig. 1b). The participants were then asked to judge whether the orientation of the probe bar was old or new. In the SO condition, the probe orientation and the stimuli were the same (i.e. old) in 50% of the trials, and different from the selected stimulus by ±4°, ±8°, ±12°, ±16°, ±20°, ±24°, ±28°, or ±32° in the remaining 50% (i.e. new). In the VO condition, the orientation of the probe was 0°, ±10°, ±20°, ±30°, ±40°, ±50°, ±60°, ±70°, ±80°, or −90° from the mean orientation, all with equal probability. After entering the response, auditory feedback was given to the participants for 500 ms only in the SO condition.

In Experiment 2, 24 participants performed a continuous estimation task. Specifically, they were presented with the 20 oriented bars for 100 ms, followed by a blank retention interval for 900 ms, and then presented with a location cue for 600 ms (Fig. 1c). This cue was randomly placed in one of the 20 stimulus locations, and indicated the orientation to be remembered. The participants were then presented with the probe bar in the center of the screen, and instructed to adjust its orientation to be as close they could get to the cued orientation. After entering the response, the correct orientation of the cued stimulus was presented for 500 ms for both SO and VO conditions as feedback.

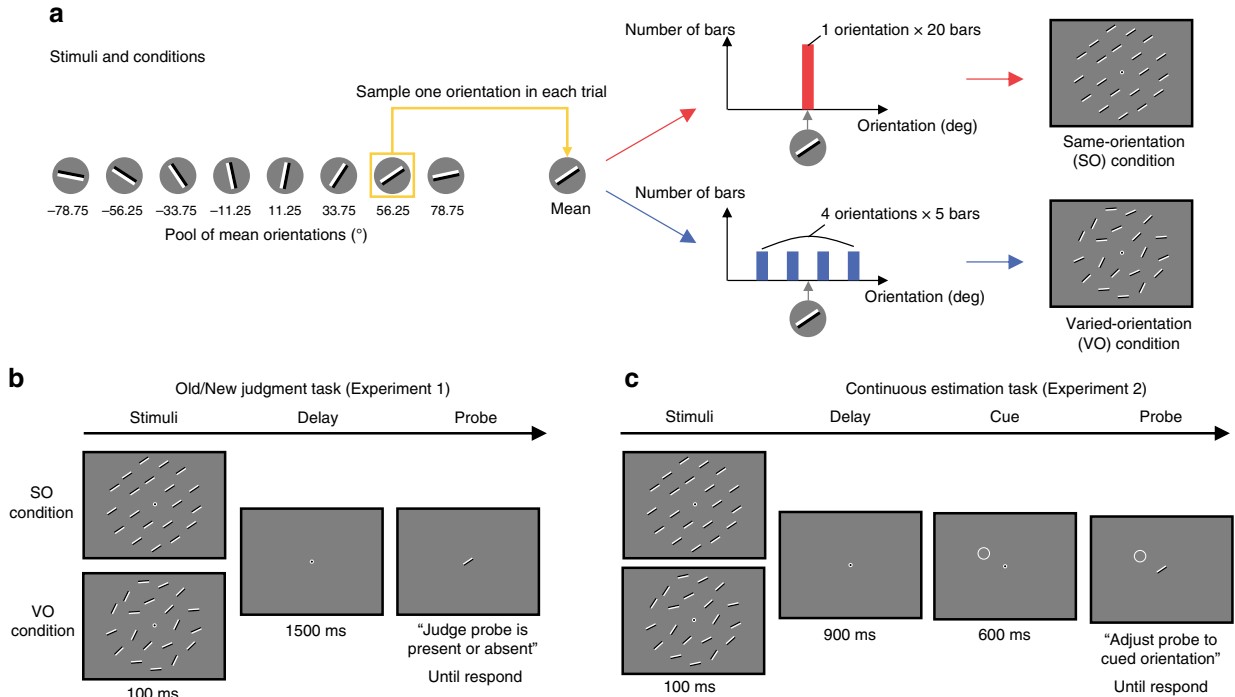

**Fig. 1 Illustrations of stimuli, conditions, and task sequences. a** Illustrations of how stimuli and conditions were created. First, one orientation was sampled from eight orientations and then used as a mean orientation in each trial of two conditions. In the same-orientation (SO) condition, the sampled orientation was identical to the orientation of 20 bars. In the varied-orientation (VO) condition, four orientations were varied around the sampled orientation, with 5 bars in each orientation, producing 20 orientations. **b** Trial sequence of the old/new judgment task in Experiment 1. Participants viewed the stimuli of the SO or VO condition for 100 ms. After a 1500 ms retention interval, the participants were asked to judge whether the probe was of a new orientation, or whether it was present in the old stimulus display. **c** Trial sequence of the continuous estimation task in Experiment 2. Participants viewed the stimuli of the SO or VO condition for 100 ms. After a 900 ms retention interval, the location cue indicated which stimulus the participants should estimate. Participants were then asked to adjust the probe orientation to the target orientation in the cued location.

The results of the SO condition in Experiment 1 showed sharp tuning of "old" responses around the target orientation (Fig. 2a). This contrasts with the results of the VO condition which showed broader tuning with "old" responses (Fig. 2b), indicating more precise representation of the orientation in the SO condition than in the VO condition. Importantly, the results of the VO condition showed that the participants reported more "old" responses as the probe orientation approached its mean (i.e., 0°), despite the fact that the mean orientation was not presented in the display of the VO condition. The proportion of "old" responses at the mean orientation was even higher than the proportion combined across all stimulus orientations [i.e., 10° and 30°; $t(19) = 3.74$, $p = .001$, paired $t$-test]. This result implies that the participant did represent the mean orientation. This result was also replicated in Experiment 2. Specifically, the precision of participants' responses was higher in the SO condition than in the VO condition (Fig. 2c), indicating that the participants represented the target orientation more precisely in the SO condition. On the other hand, we found that the participants' responses were systematically biased towards the mean orientation in the VO condition (all $p < .001$, one-sample $t$-tests; Fig. 2d). This result shows that the participants represented an ensemble representation that is distinct from the representations of individual items in a hierarchical manner[21].

**Reconstructing mnemonic representations**. To reconstruct mnemonic representations from EEG signals, we utilized an IEM. The IEM assumes that when we represent orientations, our brain is activated as a weighted sum of orientation-tuning functions; therefore, these orientation-tuning functions can be reconstructed

with EEG signals and appropriate weights. Specifically, we set a hypothetical orientation-tuning function for each orientation (Fig. 3a). This function was then convolved to orientations in all trials of the SO condition such that the orientation of a given trial corresponds to the peak response of the function (Fig. 3b). The orientation responses of a given trial and the corresponding EEG signals were then fed into the estimation of the weights (Fig. 3c). We termed the weights the "SO decoder" to emphasize that it is the orientation decoder trained in the SO condition. The SO decoder was then used to predict the mnemonic representation of an orientation in the SO condition (SO–SO prediction) and the VO condition (SO–VO prediction; Fig. 3d). The reconstructed responses are represented as a sum of channel responses in the feature space (Fig. 3e). For the SO–SO prediction, specifically, the reconstructed channel response should produce a single tuning function with its center at the orientation of the SO condition. On the other hand, the SO–VO prediction should reconstruct the representations of all items[29,30] as well as the ensemble because there was a single orientation in the SO condition which was identical to each mean orientation of the VO condition. Note that the channel responses in the VO condition should be more broadly tuned than in SO condition, and the tuning of the ensemble should be more sharply tuned than a of a set of four similar items. After we finished describing the procedures, we provided evidence that the orientation tuning reflects the ensemble rather than a set of four similar items.

We devised a statistical test to summarize the reconstructed orientation responses (Fig. 3f). We first reversed the sign of the offset by converting 22.5°, 45° and 67.5° into −22.5°, −45° and −67.5°, respectively. We then calculated the slope of the orientation responses to build a summary index, orientation

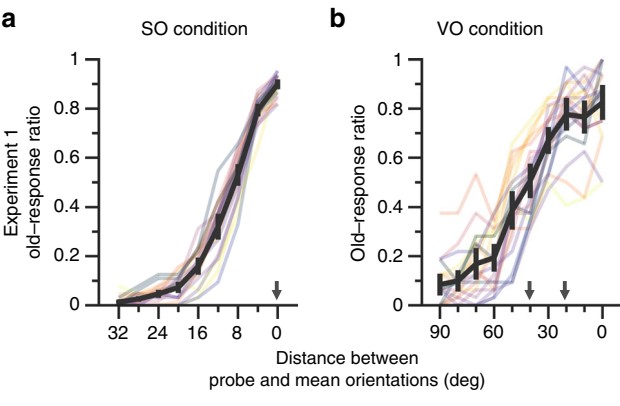

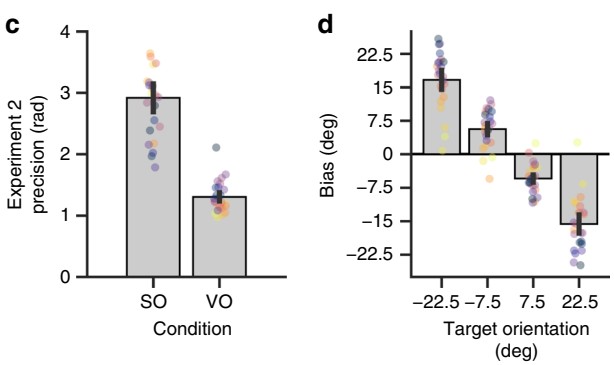

**Fig. 2 Behavioral results illustrating participants' representations of target and mean orientations. a** Participants precisely represented a single orientation in the SO condition of Experiment 1. The x-axis depicts the distance between probe orientation and stimulus orientation, and the y-axis shows the proportion of "old" responses. **b** Participants' represented mean orientations. The x-axis and y-axis are the same as Fig. 2a. **c** Participants reported orientations more precisely in the SO condition than in the VO condition. The x-axis shows conditions, and the y-axis shows mean precision of response errors. **d** Participants' response errors were systematically biased towards mean orientations. The x-axis shows target orientation, and the y axis shows the biases, calculated by mean response errors. All transparent colored lines and dots show the data of individual participants, and each color represents the same participant. All error bars represent 95% confidence intervals. SO—same orientation; VO—varied orientation.

sensitivity. A higher, negative, or zero sensitivity corresponds to sharper, reversed, or no tuning for any orientations, respectively. Statistical significance was judged using a non-parametric Monte Carlo randomization test with a significance level of 0.05. The test was uncorrected to maximize statistical sensitivity, and one-tailed to focus on non-reversed tuning functions, similar to previous studies[34,35]. We found that the sensitivities of the SO–SO predictions showed a significant increase after the stimulus onset in both Experiment 1 (Fig. 3g) and Experiment 2 (Fig. 3h). These results indicate that the SO decoder was successfully trained to decode a target orientation. Critically, we also found that the sensitivities of the SO–VO prediction significantly increased after the stimulus onset in both Experiment 1 (Fig. 3i) and Experiment 2 (Fig. 3j). These results indicate that the SO decoder successfully reconstructed the four similar orientations, as well as the mean in the VO condition.

**Temporal dynamics of mnemonic representations**. Next, we explained the temporal dynamics of mnemonic representations using a temporal generalization (TG) method. Specifically, we

trained the SO decoder at a particular time point and then generalized it to the EEG signals over the entire period. We iterated this procedure until the SO decoder trained at every time point was generalized to the EEG signals also obtained at every time point, to make a two-dimensional TG matrix of orientation sensitivities. The sensitivities of the TG matrices were then converted into a map of t-statistics using the same non-parametric Monte Carlo randomization test.

TG methods can reveal whether the representations are coded stably or dynamically over time. Stable coding refers to a pattern of neural responses over a period that stably generates the same representation over a different period. In this case, the SO decoder trained at a particular time point can reconstruct the representations of orientation over a different period, resulting in the rectangular shape sensitivity modulations in the TG matrix. Dynamic coding refers to a situation where different patterns of neural populations dynamically generate the representation over one specific period. In this case, the SO decoder trained at a particular time point can reconstruct the representation only at the matched time point, resulting in the significant sensitivity modulations along with the diagonal axis in the TG matrix. For a quantitative comparison, we developed a time-resolved stable/ dynamic index[36,37]. Specifically, the stable index is the proportion of the off-diagonal elements that were significantly greater than zero and not significantly smaller than the corresponding on-diagonal elements. The dynamic index is the proportion of the off-diagonal elements that were significantly smaller than the corresponding on-diagonal elements.

We constructed TG matrices for SO–SO and SO–VO predictions from all electrodes to ensure that we could reconstruct the orientation of the SO condition and the ensemble mean/a set of four orientations for the VO condition, and to confirm whether the stable/dynamic index effectively summarizes the patterns of two-dimensional modulations in the TG matrix. In Experiment 1, the SO–SO TG showed significant sensitivities mostly along the diagonal axis from 0 to 1,000 ms after the stimulus onset (Fig. 4a). Visual features of the SO–SO TG matrix were mirrored by the stable/dynamic index, which showed dynamic coding during an early period from 100 to 600 ms, and stable coding during a late period from 500 to 800 ms after the stimulus onset. This suggests that working memory representations of simple orientations were both dynamically and stably processed. In contrast, the SO–VO TG showed significant sensitivities over the off-diagonal areas from 300 to 800 ms after the stimulus onset, and brief dynamic coding which was dominant at an earlier period from 200 to 400 ms (Fig. 4b). In Experiment 2, we also found that both stable and dynamic coding were present in the results of both the SO–SO and SO–VO TG (Fig. 4c, d, respectively). However, compared to the results of Experiment 1, stable coding was more dominant than dynamic coding, and its above-chance sensitivities persisted. While Experiment 1 and 2 are different in several aspects, this difference at least cannot be explained by different electrode montages (see Methods and Supplementary materials). In addition, when we performed the same analysis for different frequency bands (theta- and alpha-band), spectral power appeared to hold orientation-specific representations. However, these results were spurious at best (see Supplementary materials).

**Trial-wise reconstruction of mean orientations**. One might be concerned that the SO–VO prediction reconstructs a set of four similar orientations instead of the ensemble mean. To address this issue, we conducted a trial-wise median-split analysis. Specifically, the continuous estimation task in Experiment 2 offers a rich behavioral measure than the old/new judgment. We obtained

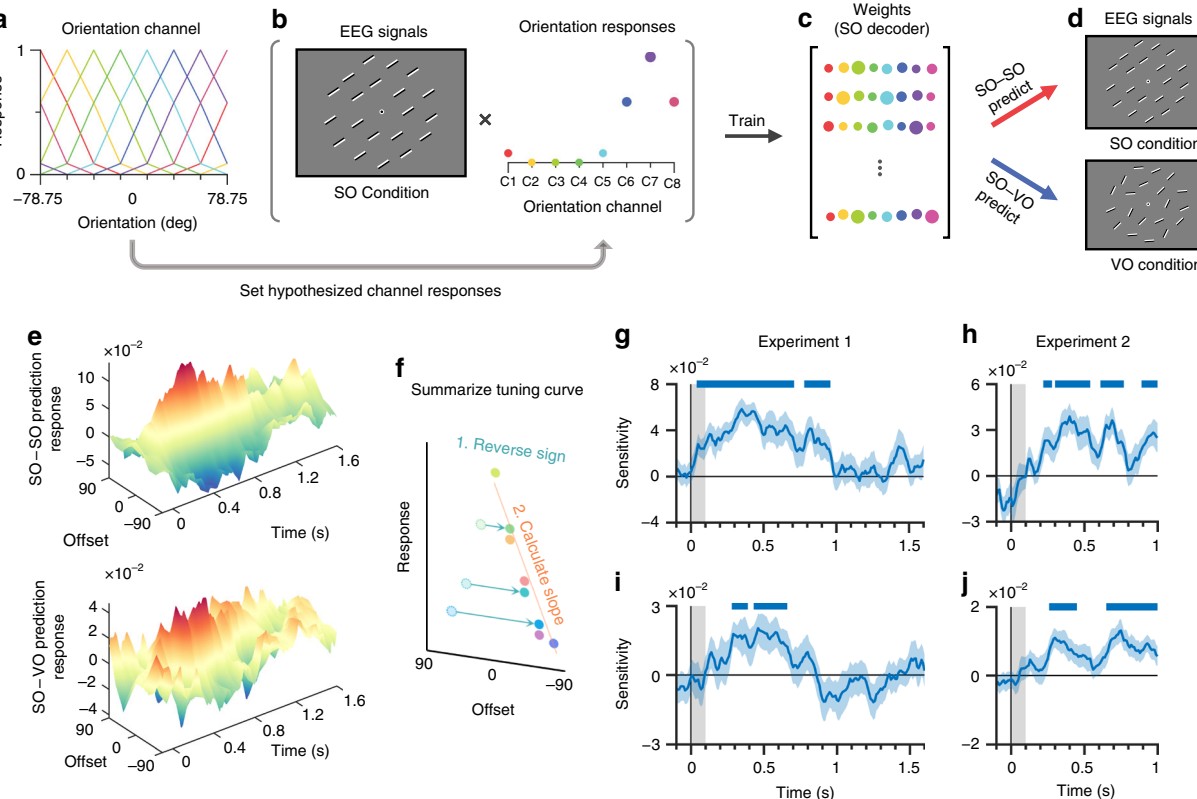

**Fig. 3 Illustrations of inverted encoding model (IEM) and procedure. a** First, hypothetical orientation-tuning functions were made for the eight orientation channels, respectively. **b** These hypothetical orientation-tuning functions were then fed into the orientation responses in each trial of the SO condition. **c** The EEG signals and orientation responses of the SO condition were used to train the weight matrix, which is termed the SO decoder. **d** The SO decoder was combined with the EEG signals of the SO and VO conditions to predict the orientation responses of each trial. The SO decoder's prediction of the orientation responses of the SO condition was termed the SO–SO prediction. SO–VO prediction refers to the SO decoder's prediction of the orientation responses of the VO condition. **e** SO–SO and SO–VO predictions showed that the reconstructed orientation-tuning functions were sharply modulated after the stimulus onset. Top, SO–SO prediction. Bottom, SO–VO prediction. **f** The orientation-tuning curves were summarized by calculating the slopes of curves. Specifically, we first reversed the sign of orientations. We then calculate the slopes by fitting the linear regression with intercept. **g–h** Orientation decoding results of SO–SO prediction during Experiment 1 (**h**) and Experiment 2 (**g**). The orientation sensitivity were sharply increased after the stimuli onset. **i–j** Orientation decoding results of SO–VO predictions during Experiment 1 (**i**) and Experiment 2 (**j**). The significant modulations showed the reconstruction of mean orientations. The x-axis shows the time after stimulus onset. The y-axis shows the orientation sensitivity. The gray rectangle from 0 to 0.1 s after the stimulus onset shows the period of the presentation of the stimuli. Blue shaded areas represent ±1 bootstrapped standard error. Blue bars above the figures (**g–j**) show significant modulation over time, determined by a one-sided Monte Carlo randomization test, $p < 0.05$. SO same orientation, VO varied orientation.

the circular median from individual participants, which lies between the mean and the target orientations (i.e., ±7.5° or ±22.5° of the VO condition). Based on the median, we separated participants' responses that were close to the cued target orientation in some trials ("target" response trials, colored blue in Fig. 5a) and to the mean orientation in the other trials ("mean" response trials, colored red in Fig. 5a). We then generalized the SO decoder to the "mean" and "target" response trials separately. If the sensitivity is higher in the "target" response trials than the "mean" response trials, we can conclude that the reconstructed orientations reflected the cued target orientation. If the reconstructed orientation sensitivity is higher in the "mean" response trials than the "target" response trials, we can conclude that the reconstructed orientations reflected the ensemble mean. Or, if the reconstructed orientation represents the four similar orientations, the variability in the reported orientations is uninformative to the ensemble or target representations; thus, we should see comparable sensitivities between the "mean" and "target" response trials.

We concluded that the SO–VO prediction did reconstruct the ensemble mean. We found that the mean orientation was stably reconstructed only from the "mean" response trials (Fig. 5b). On the other hand, orientation-specific responses were nearly absent in the "target" response trials (Fig. 5c), validating that the SO–VO prediction did reconstruct the ensemble mean rather than a set of four orientations or the cued target orientation. The different results between "target" and "mean" trials also ruled out an alternative in which participants remembered just one of the four orientations in each trial, because remembering one orientation incidentally results in a significant tuning at the mean orientation if we averaged all tunings at four orientations across all trials. The lack of "target" responses is subject to a concern that the decoder was not sensitive enough to uncover underlying representations such that participants could have held the "target" representation in a weaker strength in relative to the "mean" representation. However, this does not change the interpretation that the IEM decoded mainly the ensemble representations. Furthermore, it is unlikely that participants could have hold the cued, target orientation throughout the experiments despite the fact only one out of 20 locations, or one of four orientations was randomly selected for the report. Taken together, we concluded

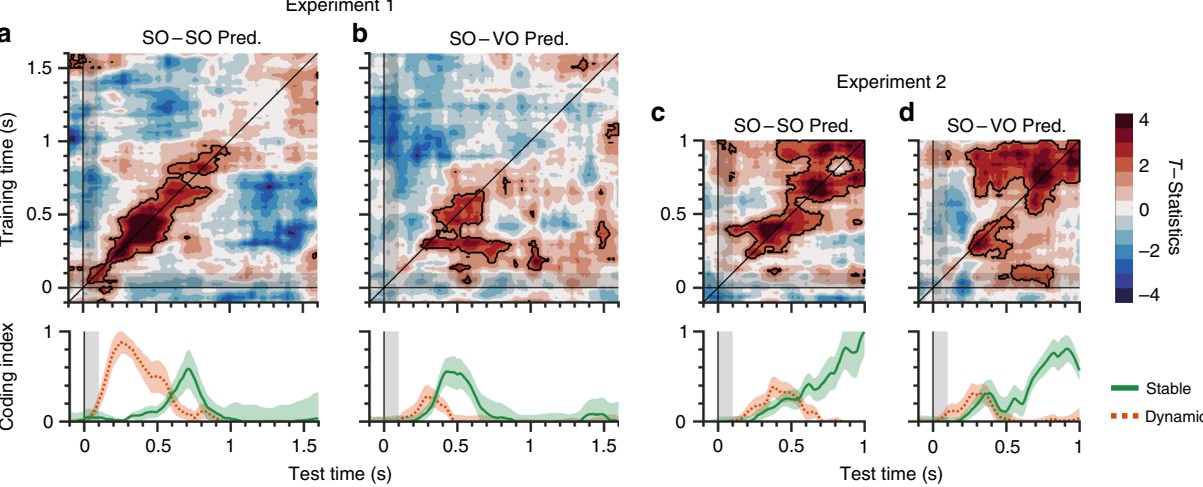

**Fig. 4 Temporal generalization (TG) methods showed distinct temporal coding, summarized by a stable/dynamic index.** First, the EEG signals at time 0 were used to train the SO decoder. This SO decoder was generalized to EEG signals over a whole period. This was repeated until the EEG signals at the final time was used to train the SO decoder. **a** SO–SO temporal generalization in Experiment 1 showed earlier dynamic coding and later stable coding, neither of which were persistent over time. **b** SO–VO temporal generalization results from Experiment 1. **c** SO–SO temporal generalization results from Experiment 2. **d** SO–VO temporal generalization results in Experiment 2. Top, the x- and y-axes show the time of test and training sets after stimulus onset, respectively. Colors represent the t-statistics. Black areas show significant modulation over time, determined by a one-sided Monte Carlo randomization test, $p <$ 0.05. Bottom, a stable/dynamic index summarized their temporal coding. The x-axis shows the time after stimulus onset. The y-axis shows the magnitude of stable and dynamic coding. Green lines show the stable coding. Red dotted lines show the dynamic coding. Shaded areas represent ±1 bootstrapped standard error. SO same orientation, VO varied orientation.

from our results that SO–VO prediction did reconstruct the ensemble mean.

**Distinct coding of frontocentral and occipitoparietal areas.** To investigate the structured representations of visual working memory distributed over different brain areas, we separated the 28 electrodes into two sets of 14 electrodes covering frontocentral and occipitoparietal areas, respectively. We then trained the SO decoder and generalized it to the SO and VO conditions from each set of electrodes. Our first observation was that the results of the frontocentral electrodes consistently showed that the stable coding was dominant in the SO–SO as well as SO–VO predictions in both experiments, while dynamic coding was nearly absent (Fig. 6a–d). On the other hand, the results of the occipitoparietal areas showed distinct patterns of results in SO–SO and SO–VO TGs and in the two experiments. Specifically, in Experiment 1, we found that the SO–SO TG showed dominant dynamic coding with modest stable coding (Fig. 6e), whereas the SO–VO TG showed modest dynamic coding but without stable coding (Fig. 6f). In contrast, in Experiment 2, we found that the SO–SO TG showed early dynamic coding and late stable coding (Fig. 6g), whereas the SO–VO TG showed a similar early dynamic coding, but the stable coding occurred at an early time period of approximately 300 ms, and persisted (Fig. 6h). These results suggest that while the working memory representations in the frontocentral areas are similar regardless of the stimulus conditions (SO vs. VO) or task specifics, the working memory representations at the occipitoparietal areas are highly task dependent, whether storing simple or abstract visual representations. These results could not be explained by either different electrode montages between the two experiments nor by eye movements (See Supplementary materials). Searchlight-based analyses also confirmed consistent patterns of results (See Supplementary materials).

**Across-participant correlation with an ensemble tendency.** In the hierarchical model of visual working memory, the degree to which a memory is biased towards an ensemble mean results from the ensemble tendency, the degree to which items are grouped in visual working memory[38]. We utilized an individual-difference approach to identify neural correlates of the ensemble tendency because it can provide another means to identify neural representation of ensembles in addition to ensemble mean. Specifically, if the SO–VO prediction does also reflect the ensemble tendency, the reconstruction from the SO–VO prediction should correlate with the degree to which responses are biased towards the mean. In the old/new judgment task of Experiment 1, the function of "old" response proportions should be narrower around the mean orientation if the participants bias their memories to an ensemble mean[38]. We thus modeled the "old" response ratios with a cumulative Weibull function in each participant and defined its threshold of reaching specific ratios as the ensemble tendency. In the continuous estimation task of Experiment 2, a stronger ensemble tendency should bias responses towards the mean to a greater extent[38–40]. We thus defined the response biases towards the mean orientation collapsed across all targets as the ensemble tendency. Finally, we summarized the neural representations of mean orientations by averaging the sensitivities within the 200–1000 ms × 200–1000 ms window in each participant.

We found that the frontocentral areas were positively correlated with the ensemble tendency in both Experiment 1 ($R^2 = 0.24$, $p = 0.03$; Fig. 7a) and Experiment 2 ($R^2 = 0.27$, $p = 0.01$; Fig. 7b). However, the occipitoparietal areas did not show any correlations in either Experiment 1 ($R^2 = 0.001$, $p = 0.91$; Fig. 7c) nor in Experiment 2 ($R^2 = 0.02$, $p = 0.56$; Fig. 7d). These patterns were consistent even when we computed the non-parametric Spearman rank-order correlation or when we defined the temporal window based on the significant sensitivities in each experiment (see Supplementary materials). These results imply

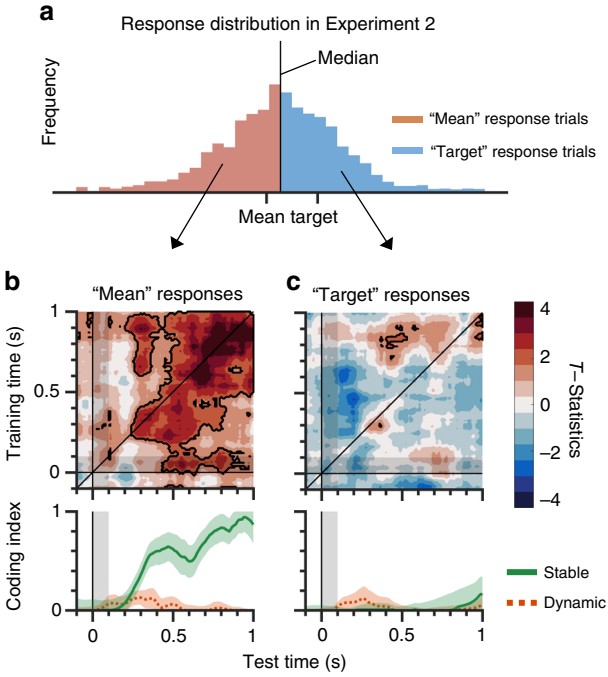

**Fig. 5 Median-split analysis from Experiment 2. a** We split a response distribution in the VO condition of Experiment 2 by its circular median. As a result, one set was closer to the mean orientation, while the other set was closer to the target orientation. The SO decoder was built and generalized to the two sets, separately. **b** Reconstruction from the mean-close responses showed stable coding. **c** Reconstruction from the target-close responses showed no evidence of either stable nor dynamic coding. Top, the x- and y-axes show the time of test and training sets after stimulus onset, respectively. Colors represent the t-statistics. Black areas show significant modulation over time, determined by a one-sided Monte Carlo randomization test, $p < 0.05$. Bottom, the x-axis shows the time after stimulus onset. The y-axis shows the magnitude of stable and dynamic coding. Green lines show stable coding. Red dotted lines show dynamic coding. Shaded areas represent ± 1 bootstrapped standard error. SO same orientation; VO varied orientation.

that the frontocentral areas, but not the occipitoparietal areas, dominantly contribute to the ensemble tendency.

## Discussion

We investigated the neural coding of structured working memory representations. Our results showed the differences in the types of representations and coding formats over the frontocentral and occipitoparietal areas. First, we reconstructed orientations of the SO condition as well as ensemble mean of the VO condition from frontocentral electrodes. This neural representation of the ensemble mean also correlated with a behavioral measure of ensemble tendency only from the frontocentral electrodes, indicating that frontocentral areas represent the abstract, ensemble mean. On the other hand, while we were able to reconstruct ensemble mean from occipitoparietal electrodes across the two experiments, they did not correlate with behaviors. Second, we consistently found stable coding in both orientations and ensemble means from the frontocentral areas, but the coding format differed over the two experiments and two conditions (i.e., SO vs. VO) in the occipitoparietal areas.

Although IEM is suitable for assessing population-level mental representations[41], IEM is methodologically limited in that it reconstructs any arbitrary, modeled channel responses[42] and the noise and population-level of neural tuning conflate the channel

responses[43]. As a result, our ability to infer tuning properties (e.g. tuning width) of the neural representations from the reconstructed channel responses is limited. Nevertheless, our results remain tenable for the following reasons. We used the permutation approach to avoid the possibility that the choice of the basis function for IEM introduced any systematic bias because we used the same basis function for generating the "null distribution." We also used the slope of the tuning function as an index of the channel sensitivity. Because the slope is an outcome of both the amplitude and width of the channel responses, our measure of sensitivity should be less sensitive to noise as well as hypothesized channel responses in contrast to the noise sensitive tuning width[43]. The fact that the reconstructed channel responses can reflect the relative likelihood of each channel makes IEM still useful in inferring relative channel responses rather than channel tuning per se. Most importantly, we built the decoder from the SO condition and applied it to different trial types of the VO condition ("mean" and "target" response trials shown in Fig. 5) to establish the ensemble representation. Together, our results remain tenable, despite the methodological limitations of IEM.

Our study contributed to literature by providing evidence that working memory representations are gradually refined such that behaviorally relevant information is abstracted over series of processing stages across the posterior-to-frontal axis[3,4,8,9]. Although we also found the ensemble mean from occipitoparietal areas, their representations should have been less refined, considering the lack of behavioral correlation. We emphasized that our experiment was different from previous studies that explicitly required participants to compute the mean of a set of stimuli[24]. Our experiment is also different from previous studies where participants were required to master category information from extensive training, which leads to similar selectivity between the PFC and the PPC[17,20]. Instead, we only asked participants to remember a set of stimuli and to determine the membership of a probe item or adjust a single cued item. The working memory should therefore have been more structured with the ensemble and individual items than in the results of previous studies[38,39].

We are nevertheless unable to rule out an alternative model focusing on what the PFC is more specialized in relative to the PPC. Xu (2018) proposed that the PFC controls and stores working memory representations and the PPC stores visual representations modulated by top-down task demands, based on the fact that the PFC also represents action[44] as well as top-down control variables[45], while the PPC represents diverse types of representations including category[19,20]. We are unable to rule out this model because we cannot evaluate action and control variables within our behavioral paradigm.

Instead, our results fit the view that the PPC stores visual representations modulated by top-down control variables, possibly guided by the PFC[46–48]. Specifically, participants were required to judge whether a probe was present or absent in Experiment 1, but the demands were different between the two conditions. In the SO condition, the probe orientation was more similar to the target orientation than in the VO condition, and feedback was given for the SO condition, but not for the VO condition. This means that participants needed to remember the orientation more precisely in the SO condition than the VO condition, even though the orientation of all 20 items was identical. In contrast, the demand for holding individual orientations should be high in Experiment 2 because participants were required to report one of them. The difference in task requirements can explain why the orientation-selective responses at the occipitoparietal areas were modest in Experiment 1 but present in Experiment 2 of the VO conditions.

Another advancement of our study is the distinguishable coding formats that we identified from TG, which would

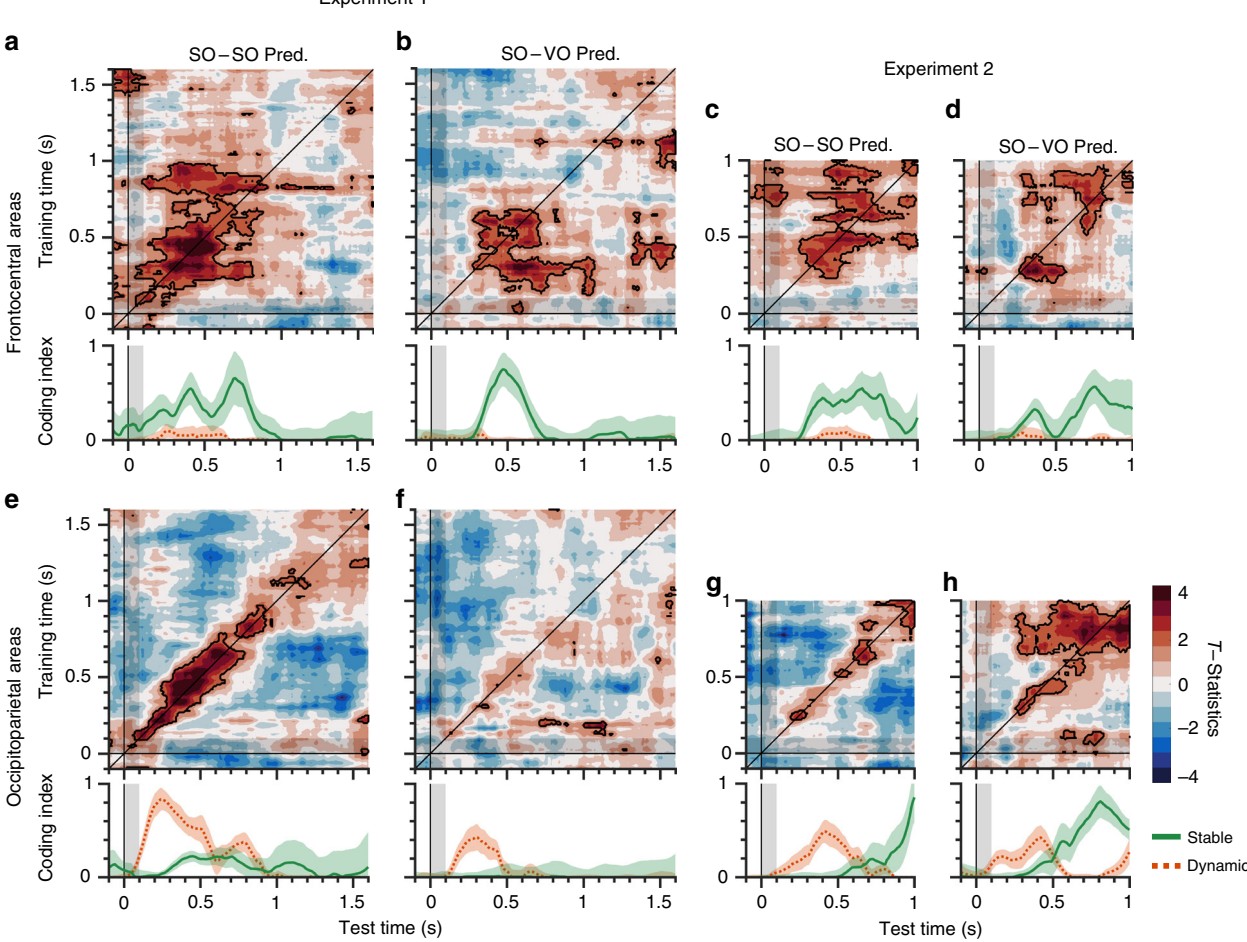

**Fig. 6 Frontocentral areas showed stable coding, while occipitoparietal areas showed stable and dynamic coding in a task-dependent manner.**
**a**–**d** Results of frontocentral areas. **a** SO–SO temporal generalization from Experiment 1. **b** SO–VO temporal generalization from Experiment 1. **c** SO–SO temporal generalization from Experiment 2. **d** SO–VO temporal generalization from Experiment 2. **e**–**h** Results of occipitoparietal areas. **e** SO–SO temporal generalization from Experiment 1. **f** SO–VO temporal generalization from Experiment 1. **g** SO–SO temporal generalization from Experiment 2. **h** SO–VO temporal generalization from Experiment 2. Top, the x- and y-axes show the time of test and training sets after stimulus onset, respectively. Colors represent the t-statistics. Black areas show significant modulation over time, determined by a one-sided Monte Carlo randomization test, $p < 0.05$. Bottom, the x-axis shows the time after stimulus onset. The y-axis shows the magnitude of stable and dynamic coding. Green lines show stable coding. Red dotted lines show dynamic coding. Shaded areas represent ±1 bootstrapped standard error. SO same orientation, VO varied orientation.

contribute to the literature in shaping the functional roles of different coding formats. Studying what information is held in different coding formats identified from EEG/MEG signals is an active research area. We found stable coding from frontocentral areas in all conditions and experiments, and this stable coding also correlated with a behavioral measure of ensemble tendency. We also found stable coding at the occipitoparietal areas in Experiment 2, where attention could have been important for reporting the cued orientation.

These results are consistent with those of recent studies, suggesting that stable coding is important in guiding our behaviors in relation to the sustained attention. Target stimulus in rapid serial visual presentation elicited category-specific stable coding only when it was subsequently reported[49]. When participants needed to determine a briefly presented target (face or house) sandwiched by masks, the participants' confidence modulated the stable coding both at the frontocentral and occipitoparietal areas[50]. In working memory literature, stable coding was modulated by selection rules at ventrolateral prefrontal regions[51] and by attentional priority at the occipitoparietal electrodes[52]. Together, these results suggest that stable coding holds task-relevant information that guides our behavior through attention or

metacognition. On the other hand, functional roles of dynamic coding remain elusive in the literature. However, considering that dynamic coding at the occipitoparietal areas represents both targets as well as distractors[49] and multiple features prior to the selection[51], it reflects a "perceptual buffer"[49,53].

We have mainly discussed the idea that the neural representations obtained from occipitoparietal electrodes originate predominantly from the PPC, since the PPC is known to store various visual representations such as orientations and categories, and it can be activated when a task requires a strong visual processing load[46,54]. Note, however, that the exact sources in the brain remain an open question due to the poor spatial resolution of EEG. In particular, we are unable to distinguish the contribution of sensory areas from the PPC obtained from the occipitoparietal electrodes, considering that mounting evidence has shown that early sensory areas including the primary visual cortex do show stimulus-specific responses over the retention interval of visual working memory tasks[55,56]. Nevertheless, EEG signals might be more suitable for providing a macroscopic view by identifying populations of neural activities originating from multiple brain networks that serve for building working memory representations and eventually guiding our behaviors, as shown

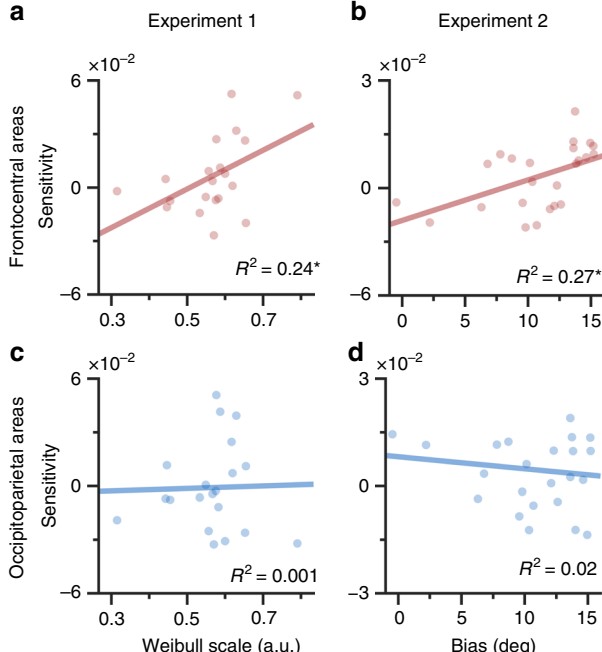

**Fig. 7 Frontocentral areas showed significant correlations between reconstructions and ensemble behavior, while occipitoparietal areas showed no correlations. a–b** Orientation sensitivities of SO–VO predictions in frontocentral areas showed correlation with ensemble behavior. **a** Results of Experiment 1. Behavioral index of Experiment 1 was calculated by fitting a cumulative Weibull model to the results of the VO condition and getting a threshold parameter. **b** Results of Experiment 2. Behavioral index of Experiment 2 was the average biases of four target orientations. **c–d** Orientation sensitivities of SO–VO predictions in occipitoparietal areas did not show correlation with ensemble behavior. **c** Results of Experiment 1. **d** Results of Experiment 2. Each data point represents the data of one participant. The x-axis shows behavioral index, and the y-axis shows the orientation sensitivities. Lines show the regression line. *$p < 0.05$, **$p < 0.01$. SO same orientation, VO varied orientation.

by the consistent stable coding from the frontocentral areas and its behavioral correlation, despite the fact that neurophysiological studies have shown varied coding formats[32].

## Methods
Twenty-seven participants including the third author (12 females; mean age ± SD = 24 ± 4.15 years) and 35 participants (20 females; mean age ± SD = 23.03 ± 2.5 years) were recruited for Experiment 1 and 2, respectively. All participants reported normal or corrected-to-normal vision and gave informed consent, approved by the Sungkyunkwan University Institutional Review Board. Seven participants were excluded from the analysis of Experiment 1 (one participant did not complete the experiment, and six participants had <80% of trials artifact-free trials). Eleven participants were excluded from the analysis of Experiment 2. Six participants had <80% artifact-free trials, and five participants could not complete the experiments due to problems with the recording system (four participants due to excessive noise in the EEG signal and 1 due to a recording failure). For 1 participant in Experiment 2, the O1 electrode was interpolated in EEGLAB due to noise in the recording. Out of the participants included in the results, only two participants completed both experiments. The sample size was closely matched to previous studies that used similar methods[52,57].

**Behavioral protocols**. We conducted an old/new judgment task in Experiment 1 and a continuous estimation task in Experiment 2. The numbers of trials in the SO and VO conditions were 1024 and 288 (Experiment 1) and 512 and 512 (Experiment 2), respectively.

**Experiment 1**. Stimuli were generated and controlled using a Mac Mini, MATLAB, and the Psychophysics Toolbox[58,59]. The stimuli were presented on a CRT monitor with a refresh rate of 100 Hz and a resolution of 1,024 × 768 pixels at a viewing

distance of ~85 cm. A gray background and a black fixation point with a white outline (0.256°) were maintained throughout the experiment. Twenty oriented bars (1.07° × 0.256°) were displayed for the memory items. We prepared two bar types in opposite polarity to equate their mean luminance to the background and displayed them in equal numbers. In each trial, the mean orientation of the 20 oriented bars was randomly determined out of −78.75°, −56.25°, −33.75°, −11.25°, 11.25°, 33.75°, 56.25°, and 78.75°. The distribution of the orientations was chosen according to two conditions: SO and VO. In the SO condition, the orientations of the 20 bars were all the same as the mean orientation; in the VO condition, the orientation of the 20 bars were varied between −30°, −10°, +10°, and +30° from the mean orientation, with 5 repetitions of each. The 20 bars were placed around the circumference of two invisible circles. Specifically, seven and thirteen oriented bars were placed around an inner circle (radius of 2.2°) and an outer circle (radius of 4.09°), respectively. Thus, any three bars in the closest neighborhood were placed at an approximately equal distance. To add location randomness, the phase of each circle was randomized, and jitter was added to each location with a distance of 0–0.41° randomly for all directions.

The probe bar was identical to the oriented bars in shape and located at the center of the screen. In the SO condition, the orientation of the probe was identical to the orientation of the memory items in 50% of trials. In the remaining 50% of trials, the orientation of the probe was ±4°, ±8°, ±12°, ±16°, ±20°, ±24°, ±28°, or ±32° from the orientation of the memory items. In the VO condition, the orientation of the probe was 0°, ±10°, ±20°, ±30°, ±40°, ±50°, ±60°, ±70°, ±80°, or −90° from the mean orientation of the memory items, with equal probabilities. We made the range of probe orientations of the SO condition narrower than that of the VO condition so that participants remembered the orientations precisely. The polarity of the probe was randomly determined for each trial.

Each trial began with a fixation point. After 700–1000 ms, 20 oriented bars were displayed for 100 ms, after which they disappeared. After 1500 ms, a probe bar was displayed, and the participants judged whether the probe had been presented before (old/new judgment). Only in the SO condition, a low or high frequency feedback sound was played to indicate that the response was correct or incorrect, respectively. The next trial began 1000 ms after the response. Participants completed 32 blocks of the SO condition and 32 blocks of the VO condition, and they completed 32 trials (1024 trials in total) in each block of the SO condition and nine trials per block (288 trials in total) in the VO condition. The blocks of the SO and VO conditions were presented alternately. All participants completed a practice block of 24 trials in advance. The condition practiced in the initial block was counterbalanced across participants. The experiment consisted of 1312 trials, and took participants approximately 3.5 h to complete from electrophysiology preparation to hair cleaning.

We quantified the degree of ensemble representation from the behavioral responses in the VO condition. We obtained a measure of how similar a probe was to the mean orientation in the following way:

$$\text{sim}(\theta) = 1 - \frac{|\theta|}{\pi/2},$$

where $\theta$ is the probe orientation relative to the mean orientation of the stimulus in radians, and sim is the orientation similarity to the mean orientation. We then fitted a cumulative Weibull function:

$$W(\text{sim}; \alpha, \beta, \gamma, \delta) = \gamma - (\gamma - \delta) \cdot e^{-(\text{sim}/\alpha)^\beta},$$

where sim is the orientation similarity, $\alpha$ is the threshold at which a response function reaches 63.2% of its full amplitude, $\beta$ is the slope, $\gamma$ is the upper horizontal asymptote, and $\delta$ is the lower horizontal asymptote of a response function. We used a maximum likelihood estimation (MLE) with the joint probability mass function for the fitting procedure:

$$f(k|n, \text{sim}, \alpha, \beta, \gamma, \delta) = \prod_{i \in \text{sim}} \frac{n_i!}{(n_i - k_i)! k_i!} W(\text{sim}; \alpha, \beta, \gamma, \delta)^{k_i} (1 - W(\text{sim}; \alpha, \beta, \gamma, \delta))^{n_i - k_i},$$

where $n$ and $k$ are the number of trials and "old" responses in similarity $i$, respectively. The goal of the MLE was to find the set of parameters that minimizes the negative log-likelihood function:

$$-\ln L(\alpha, \beta, \gamma, \delta|n, k, \text{sim})$$
$$= -\sum_{i \in \text{sim}} [k_i \cdot \ln W(\text{sim}; \alpha, \beta, \gamma, \delta) + (n_i - k_i) \cdot \ln\{1 - W(\text{sim}; \alpha, \beta, \gamma, \delta)\}].$$

The fitting was completed using the fmincon function in MATLAB with boundaries of $\alpha$ (0, 1), $\beta$ (0, ∞), $\gamma$ (0, 1), and $\delta$ (0, 1) to constrain the parameters. To maximize the probability of finding a global minimum, the initial seeds were randomly selected in $\alpha$ (0, 1), $\beta$ (0, 10), $\gamma$ (0, 1), and $\delta$ (0, 1). Finally, after 30 repetitions, the parameters showing the minimum negative log-likelihood were used for selecting the parameters. Of the four parameters, the threshold parameter $\alpha$ was used for further analyses because the threshold is closely related to the narrowness of a response function, and thus the degree of ensemble representation.

**Experiment 2**. All aspects were identical to Experiment 1 except the following changes. The four orientations of the VO condition were −22.5°, −7.5°, +7.5°, and +22.5° from the mean orientation. The diameter of a location cue was 1.07°. Participants viewed a location cue for 600 ms after a 900 ms retention interval and

   9

adjusted a bar in the test phase to the cued orientation. We reduced the retention interval from 1500 ms in Experiment 1 to 900 ms in Experiment 2 so that participants could finish the entire experiment within a reasonable time, as the adjustment took longer than the old/new judgment. The orientation of the probe bar was randomly selected for each trial. The actual orientation of the cued bar was given immediately after the response and remained on the screen as feedback for 500 ms in all trials. There were 512 trials in both the SO and VO conditions with 16 in each block. The 4 orientations used in the VO condition were probed equally as often. Participants completed 1024 trials in approximately 3.5 h from electrophysiology preparation to hair cleaning.

**EEG acquisition**. The EEG signal was recorded at a sampling rate of 500 Hz using 32 Ag/AgCI electrodes mounted in an elastic cap and amplified using an Acti-CHamp amplifier (BrainVision). The signal was low-pass filtered (140 Hz) online. The 28 electrode sites consisted of Fz, F3, F4, FC1, FC2, FC5, FC6, Cz, C3, C4, T7, T8, CP1, CP2, CP5, CP6, Pz, P3, P4, P7, P8, POz, PO3, PO4, PO7, PO8, O1, and O2 (Experiment 1), or Fp1, Fp2, Fz, F3, F4, F7, F8, FC1, FC2, FC5, FC6, Cz, C3, C4, T7, T8, CP1, CP2, CP5, CP6, Pz, P3, P4, P7, P8, Oz, O1, and O2 (Experiment 2). All electrodes were recorded using the left mastoid as a reference online and re-referenced by the average of the left and right mastoids offline. To detect blinks and eye movements, we recorded vertical electrooculography (EOG) from below the left eye and horizontal EOG from the external right ocular canthus. The impedance of all electrodes was kept below 10 kΩ throughout the experiment.

**EEG preprocessing**. EEG signals were preprocessed using MATLAB with the EEGLAB[60] and ERPLAB[61] toolboxes. A non-causal infinite impulse response (IIR) Butterworth high-pass filter (−6 dB half-amplitude cutoff frequency of 0.1 Hz and 12 dB/oct roll-off) was applied to the continuous EEG signals using pop_basicfilter.m in ERPLAB. Epochs of the signals were taken from −200 ms to 1600 ms (Experiment 1) and from −200 ms to 1000 ms (Experiment 2) relative to the onset of the memory items. Each epoch was checked by visual inspection for blinks, eye movements, and nonstereotyped artifacts. Trials with artifacts were excluded from the analysis. The EEG data were then smoothed using a 51-point moving-average filter (i.e., 100 ms sliding window) and down-sampled to 100 Hz to enhance the signal-to-noise ratio and for computational efficiency, similar to previous studies[52,62].

**Inverted encoding model**. We applied an IEM with three-fold cross-validation using 100 iterations. When we separately analyzed frontocentral and occipitoparietal areas, the frontocentral electrodes consisted of Fz, F3, F4, FC1, FC2, FC5, FC6, Cz, C3, C4, T7, T8, CP5, and CP6 in Experiment 1 and Fp1, Fp2, Fz, F3, F4, F7, F7, FC1, FC2, FC5, FC6, Cz, C3, and C4 in Experiment 2. The occipitoparietal electrodes consisted of CP5, CP6, Pz, P3, P4, P7, P8, Poz, PO3, PO4, PO7, PO8, O1, and O2 in Experiment 1 and T7, T8, CP1, CP2, CP5, CP6, Pz, P3, P4, P7, P8, Oz, O1, and O2 in Experiment 2.

The IEM was used to reconstruct the remembered orientation. The IEM assumes that the brain signal of each electrode is the weighted sum of an orientation-selective tuning function. This is characterized by a general linear model of the following form:

$$B_{(m \times n)} = W_{(m \times k)} C_{(k \times n)}.$$

Specifically, $B$ ($m$ electrodes × $n$ trials) is an obtained brain signal and $W$ ($m$ electrodes × $k$ orientations) is an arbitrary weight matrix, mapping from the orientation-selective tuning function to the EEG signal. Lastly, $C$ ($k$ orientations × $n$ trials) is a channel tuning function (CTF) matrix, reflecting an orientation-selective tuning function in each trial.

We first modeled the basis function of the orientation response profiles. We assumed that the orientation-selective tuning function should have the following form:

$$R = \cos^7 \theta,$$

where $R$ is the response of orientation channels in arbitrary units, and $\theta$ is an orientation angle (i.e., −90°, −67.5°, −45°, −22.5°, 0°, 22.5°, 45°, 67.5°) in radians. With this basis function, we constructed the CTF matrix for each target orientation that corresponded to the presented orientation of a given trial of the SO condition.

Next, we applied three-fold cross-validation with 100 iterations. Specifically, we partitioned each participant's artifact-free trials into three subsets without replacement. The partition was random with the constraint that the number of trials in each target orientation and each subset was the same (any trials exceeding the number were discarded). In each subset, the trials of each target orientation were averaged. We then treated two subsets as a training set ($B_1$ and $C_1$) and the remained subsets as a test set ($B_2$ and $C_2$). We then obtained the estimate of the weight matrix ($\hat{W}$) with the training set using the least squares method:

$$\hat{W} = B_1 C_1^{\mathrm{T}} \left( C_1 C_1^{\mathrm{T}} \right)^{-1}.$$

The CTF matrix of the test set ($\widehat{C_2}$) was then obtained with the estimated weight matrix using the following formula

$$\widehat{C_2} = \left( \hat{W}^{\mathrm{T}} \hat{W} \right)^{-1} \hat{W}^{\mathrm{T}} B_2.$$

This procedure was repeated until all three sets were tested, and the whole procedure was iterated 100 times with a new random partition. The CTF matrix was averaged across all folds and iterations. We corrected the baseline CTF of each target orientation and time by subtracting the mean of channel responses of all orientations from the channel response of each orientation. The CTFs in each target orientation were then circularly shifted to a common center and all eight aligned CTFs were averaged.

To summarize the decoding sensitivity, we calculated the linear slope of the CTFs. Specifically, we reversed the signs of the orientation channels 22.5°, 45°, and 67.5° to become −22.5°, −45°, and −67.5°, respectively. We then fitted the linear regression with an intercept across the eight channel responses (i.e., −90°, −67.5°, −67.5°, −45°, −45°, −22.5°, −22.5°, 0°) and obtained the linear slope of the CTFs. When doing so, zero decoding sensitivity corresponds to no orientation sensitivity, while higher decoding sensitivity corresponds to greater orientation sensitivity.

**TG of the IEM**. To assess the temporal generalizability of the CTFs, we implemented the TG of the IEM. Specifically, we trained a weight matrix from the training set at time $t$ and applied the estimated weight matrix to the test set at time $t'$. This procedure was repeated so that the weight matrices at every time point had been used to calculate the slope of the CTFs (decoding sensitivity) at every time point, thereby creating a two-dimensional TG matrix of the CTF slopes. All other aspects (e.g., three-fold cross-validation with 100 iterations) were identical to the IEM procedure explained above.

**Stable/dynamic index**. To quantify the magnitude of stable and dynamic coding, we devised a stable/dynamic index[36,37]. We first sought to test whether there was significant dynamic or stable coding at a particular time point. For the dynamic coding, we tested whether the off-diagonal element of the matrix TG($t_1,t_2$) was significantly smaller than the corresponding on-diagonal elements of the matrix TG($t_1,t_1$) and TG($t_1,t_2$)

$$H_1(t_1, t_2) = \mathrm{TG}(t_1, t_2) < \mathrm{TG}(t_1, t_1)$$
$$H_2(t_1, t_2) = \mathrm{TG}(t_1, t_2) < \mathrm{TG}(t_2, t_2)$$
$$\mathrm{dynamic}(t_1, t_2) = H_1(t_1, t_2) \wedge H_2(t_1, t_2).$$

For the stable coding, we tested whether the off-diagonal element of the matrix TG ($t_1,t_2$) was significantly higher than zero and, at the same time, not significantly smaller than the corresponding on-diagonal elements of the matrix TG($t_1,t_1$) and TG($t_2,t_2$):

$$H_3(t_1, t_2) = \mathrm{TG}(t_1, t_2) > 0$$
$$\mathrm{stable}(t_1, t_2) = \neg H_1(t_1, t_2) \wedge \neg H_2(t_1, t_2) \wedge H_3(t_1, t_2).$$

We applied these tests to all off-diagonal elements of the matrix by using permutation tests.

We then summarized temporal dynamics over time. We gained the indices of stable or dynamic coding at a time point $t$ by calculating the proportion of significant stable or dynamic coding elements over a 310 ms square window centered around time point $t$. We simultaneously controlled the temporal smearing effect caused by a 100 ms moving-average filter by excluding the elements within ±50 ms of the diagonal axis from the analysis. A stable or dynamic index of 1 indicates that the TG is completely stable or completely dynamic, whereas 0 indicates that the TG is not at all stable or dynamic.

**Cross-conditional generalization of the IEM**. To find the neural correlates of ensemble representation, we examined the cross-conditional generalization from the SO condition to the VO condition. Specifically, we implemented the TG to the SO condition as the training set and the VO condition as the test set. When applying the three-fold cross-validation with 100 iterations, we randomly partitioned the SO condition into three subsets, trained the weight matrix with two subsets, and applied the weight matrix to all trials of the VO condition. We repeated this procedure until all subsets of the SO condition were equally used for training and iterated 100 times with a new random partition. All other aspects (e.g., TG) were identical to the previously explained IEM procedures.

**Permutation test**. To investigate whether the decoding sensitivity was significantly above that which could be expected by chance, we tested whether it was greater than zero using one-sample $t$-tests. Furthermore, because the decoding sensitivity could not have been normally distributed, we utilized a permutation test to approximate the null distribution of the $t$-statistics[57]. Specifically, we first randomly shuffled the orientation label of all trials in the SO and VO conditions and then fitted the IEM. With 1000 iterations, we obtained a null distribution and then calculated the $t$-statistic probabilities. Our permutation test was one-tailed, and $p < 0.05$ was considered statistically significant.

**Bootstrapping method**. For visualizing the standard error of the decoding sensitivity and stable/dynamic index, we implemented a bootstrapping method. From the whole dataset including all participants, we produced 10,000 bootstrapped samples by drawing the number of participants with replacement, and then calculated the 10,000 means of the bootstrapped samples. We obtained the standard errors by calculating the standard deviation of those 10,000 means. Note that when bootstrapping the stably/dynamic indices and calculating standard errors, we utilized significance maps (i.e., H1, H2, and H3) that are obtained by parametric $t$-tests, instead of by Monte Carlo randomization tests, for computational efficiency. The stable/dynamic indices of parametric t-tests were only slightly different from that of randomization tests (less than 0.01 on average).

**Reporting summary**. Further information on research design is available in the Nature Research Reporting Summary linked to this article.

## Data availability

Data used in this study are available in [https://doi.org/10.17605/OSF.IO/NVAZ3]. A reporting summary for this Article is available as a Supplementary Information file.

## Code availability

The code used in this study are available in [https://doi.org/10.17605/OSF.IO/NVAZ3]

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

## Acknowledgements
This research was supported by the Brain Research Program through the National Research Foundation of Korea (NRF) funded by the Ministry of Science and ICT (NRF-2017M3C7A1029658) and by National Research Foundation (NRF-2016R1D1A1B03 930292, NRF-2019R1A2C1005978) to M.K., and by the Institute for Basic Science (IBS-R001-D1) to Y.K. The authors thank S.C. Chong, K.-J. Tark, W.M. Shim for discussions.

## Author contributions
Y.K. and M.K. conceived the study; B.O. and M.K. designed the experiments. B.O. performed research, analyzed the data, and wrote a draft. All authors discussed the results and contributed toward the manuscript.

## Competing interests
The authors declare no competing interests.
