## [Peer Review File · Nature Communications]

Reviewers' Comments:

Reviewer #1:

Remarks to the Author:

Oh et al. tested the hypothesis that the PFC and PPC are using different encoding schemes in visual working memory task. This was done by applying the decoding algorithm on the broadband scalp EEG signal from the electrodes divided in the anterior and posterior set. While the question of different encoding schemes applied in different modules has relevance for mechanistic understanding of visual working memory, multiple problems with methodology and presentation make the interpretation of results questionable.

Major comments:

- 1) The key parameters of assessing the decoder are missing in this paper, such as sensitivity, specificity and ROC and its AUC. The authors should show the summary plot of tuning curves, which should reveal if there is a spatial clustering of electrodes highly contributing to decoding. In addition, the assumption that the signal on posterior electrodes could be specifically ascribed to PPC, and not to a variety of visual system areas is questionable. The absence of more precise source localization makes the anatomical claims less valid.
- 2) The results failed to challenge the view that PFC and PPC distinction is based on the level of abstraction (Line 303). The basic sensory information (orientation) and its ensemble representation (i.e. orientation mean) could be encoded in both the homogeneous and heterogeneous during the retention period, after the subjects were trained well to retain the mean orientation in order to get correct feedback afterwards. As a result, it is hard to differentiate whether the information encoded in the anterior region pertains to orientation or ensemble representation or is it the epiphenomena of training.
- 3) We are not getting any sense on what the stable coding looks like in the EEG signal, except that it contains the information sufficient for decoding at better than chance level. Is it a persistent oscillation in certain frequency range, consistent phase offset between the signals on different electrodes, some other signal feature(s) or feature combination?
- 4) All of the key terms, including the dynamic and stable population coding, should be unambiguously defined at the outset. Considering the breadth of NatComm readership, the terms such as 'ensemble representation' - in this context used to describe the neural coding of higher order statistics from the set of visual stimuli - could be understood by many readers as denoting the stimulus representation at neural ensemble level. In addition, usage of different labels for the same concepts makes the paper difficult to follow. For example, 'homogenous' and 'inhomogeneous' trials are in some instances labeled as 'orientation' and 'ensemble'.
- 5) If the hypothesis pertains to differential encoding schemes in PFC and PPC, why was the decoding on Fig. 2 and 3 done on the electrode set that included both anterior and posterior electrodes?
- 6) It seems problematic to assume that the short autocorrelation time constant of broadband EEG (referred to as 'stable coding') implies that the 'same neural population represents the target for that period' (Lines 74-76). For example, different subsets of hippocampal place cells could encode different parts of the environment, while the theta rhythm in the local EEG would be stable.

Minor comments:

- 1) Figure captions should start with a headline describing the results, not just describing which variables are shown on the figures.
- 2) The authors claim to be using the 10-20 electrode placement system, which doesn't seem to be the case, based on the electrode labels.
- 3) Fig.1. A and C. It would help to put the arrow that denotes the sequential order of different trial epochs. Also, the meaning of orange and purple color codes is not denoted in the figure caption.

Reviewer #2:

Remarks to the Author:

This paper explores whether working memory representations are neurally coded in a stable or dynamic format. The authors use an ensemble paradigm, whereby observers had to maintain an average orientation in working memory. Because ensembles are represented hierarchically, this allowed for the exploration of working memory stability at multiple stages of processing. In two experiments, observers viewed homogeneous and heterogeneous sets of oriented stimuli and had to decide whether a test stimulus was a member of the preceding set (Experiment 1) or adjust a test stimulus to match the orientation of a cued set member (Experiment 2). The authors trained a decoder on the homogeneous condition to identify the viewed orientation based on a weighted combination of EEG electrode site activation. They then used TG matrices to determine how effective the decoder was in identifying the viewed orientation at various time points when trained on one specific time point. Finally, they tested how training on the homogeneous set generalized to the heterogeneous sets for both anterior and posterior electrode sites. Overall, authors claim the decoder showed dynamic coding for homogeneous sets and relatively stable coding for heterogeneous sets. When broken down by electrode sites, anterior regions (what authors claim to reflect PFC) were more stable for both homogeneous and heterogeneous sets while posterior regions (PPC) were more dynamic (although the results for posterior were somewhat inconsistent). Authors cite these results as evidence for differential working memory representation structures across different regions of the brain.

These experiments demonstrate a sophisticated and clever approach to exploring the nature of working memory representations. I also commend the authors on their validation technique for determining whether observers were actually representing the average orientation and not just a member of the set. Admittedly, I am not in a position to evaluate whether the decoder was implemented correctly, but I can evaluate the logic of the approach along with whether the interpretation has merit. Here are some of my concerns:

I am somewhat confused about how the results of the TG matrices, which showed either dynamic or stable ensemble working memory representations, are to be interpreted. What does it mean to have a dynamic working memory representation if that representation is unchanging for the duration of a trial? Why would I have to recruit different neural networks to maintain an unchanging item? Are there advantages to one system being stable as opposed to dynamic? I think some of these concerns can be addressed in an elaborated introduction section, which was somewhat lacking in its breadth (and sometimes confusing, especially the last paragraph). Many of the concepts introduced are insufficiently described, and if authors want to reach a wider audience, care should be taken to more fully describe these challenging methodological approaches.

Discussion:

Why does it follow that stable coding vs. dynamic coding for ensembles indicates task general vs. task specificity? What does it mean to have an ensemble dynamically coded, if the representation doesn't change over the time observers are maintaining it?

The explanation for the 'reconfiguration' of PPC depends on the homogeneous task being more challenging than the heterogeneous task, correct? Is there behavioral evidence to support this? In almost all other research comparing homogeneous and heterogeneous performance, observers do better when the set is homogeneous (sometimes a set size effect emerges where heterogeneous gets slightly better, but these are uncommon and hard to replicate).

Beware of making causal inferences from the decoding data (page 16, line 337)

Methods:

There is an unusually large number of participant exclusions (nearly a third of your sample). This is not typical of EEG studies, so further justification for this is warranted.

These experimental sessions were exceptionally long (3.5 hours). How did authors ensure participants were compliant and attentive throughout this time?

I don't quite understand the logic of how making the probe ranges narrower in homogeneous trials makes the orientations more memorable. Please further explain this.

The circular median isn't sufficiently explained. Is this just accounting for the fact that orientation is a circular space (i.e., 1° is actually close to 179°)? Perhaps more importantly, can authors please elaborate on what the median split is designed to achieve?

There are many decision points in the the implementation of the IEM. Citations and further justification for these decisions are warranted.

Perhaps this is implicit in the IEM implementation, but how can authors be so certain of the source of their anterior/posterior electrode activations to the point that they claim their results are primarily driven by differences between the PFC and PPC?

Other concerns:

The task as described in Figure 1 is insufficient. What is the old/new judgment based on? This becomes clearer in the text, but the figures and their captions should stand on their own.

How were the incremental steps of the test stimulus selected? Was there pilot testing?

In the continuous response experiment, the subjects were instructed to adjust the stimulus to match one of the 4 items presented, correct? This is not entirely obvious in the text, currently.

It might be helpful to include a legend in Figure 5.

Jason Haberman

Reviewer #3:

Remarks to the Author:

The authors present a clever experiment investigating the neural basis of ensemble representations in memory. The encoding/decoding approach is very interesting and the design is novel. There are concerns about the stimulus choice, the comparison between homogeneous and heterogeneous, and whether the results reflect a true ensemble representation. More quantitative comparisons between conditions would also help. With revision, the manuscript will be a great addition to the literature.

Fig. 1 Where's the data for the homogeneous condition? It's important to compare the homogeneous to the heterogeneous condition.

How do we know observers don't just sample one gabor patch in the heterogeneous condition? If observers picked one gabor patch and based decision on that (old/new or estimation/matching/adjustment), their data would look similar to that in Fig 1. On average, across trials, sampling single Gabor patches would produce data that is similar to that in Fig. 1. In other words, what is the integration efficiency?

Fig 1D. Are the light colored dots individual subjects? The figure caption is unclear. It says "individual data." What is "individual data"? Individual subjects? Sessions? Individual blocks? Individual trials (too few to be that, I assume, but it's unclear).

Fig. 2A. Does "orientation" refer to only the homogeneous condition? The mapping of homogeneous and heterogeneous onto the "orientation" and "ensemble" is unclear in Fig 2.

Fig. 2B. If this same analysis were done for all orientations around the mean (not just the mean), what would decoding performance look like? Is decoding best specifically for the mean, or are all orientations in the heterogeneous display decoded just as well? If the decoding is sharply tuned to average orientation, that would be nice to know.

Fig. 4. Why separate electrodes in this way? Seems arbitrary. The comparison of temporal tuning for the different arrays of electrodes also seems descriptive rather than quantitative. Many of the results are presented as visual comparisons of heatmaps, which is not ideal.

Line 236-238. The statements here don't seem well supported by the data, or at least are vague descriptions without any direct statistical comparison. E.g., "...spread out broadly across the horizontal axis..." Everything is spread out broadly across the horizontal axis... We need a more precise metric of the tuning and a direct comparison between panels. We also need a qualification that this is based on a very arbitrary division of electrodes in anterior and posterior.

Was there source localization? How do we know the electrodes genuinely reflect anterior and posterior cortical activity?

The cue is unclear in methods and in main text. Was the cue always in the same spatial location? There was always only one cue?

If the cue was in a predictable location, it seems that subjects could attend to that location, and probably simply maintain and report that single orientation. It would be very surprising to find bias in reported orientation for an oriented Gabor patch in a known location, unless there was crowding. But perhaps the cue location is randomized in some way.

Assuming the cue was in an unpredictable location, then the question is whether subjects actually

integrated more than one gabor patch or just relied on one oriented stimulus to make their judgments. Across trials it would look like an ensemble, once data is averaged, but on any single trial it might be a single oriented Gabor patch that drives responses. The data in Fig 1D do not address this. Those colored dots represent averaged data, so on any trial subjects may have used just one stimulus. There may be a way to simulate performance or analytically prove that observers used multiple stimuli in their ensemble estimates. This is foundational for all of the results

The brain could use ensemble representations for both heterogeneous and homogeneous sets of stimuli. The difference is really the variance of the display, not necessarily whether an ensemble is used. The IEM is very clever, and it may make sense to use homogeneous (or perhaps better to use single stimuli) to build that decoder. However, when the data is presented in the figures and discussed, it is presented by comparing "orientation" versus "ensemble" panels. Yet, homogeneous and heterogeneous sets of stimuli may be represented as ensembles, so the orientation data (e.g., Fig 2A,C, E,G) may reflect ensemble representations. Any difference is really about the variance in the display. In fact, homogeneous displays may be better represented/encoded/maintained as an ensemble. Increasing variance tends to reduce sensitivity to ensemble characteristics in many domains. One useful approach would be to investigate how much orientation variance needs to be added to the display to change the data in Fig. 1E toward Fig. 1F.

"...The dominant contribution of anterior brain regions to the ensemble representations." Specify that this about the "maintained" or "recalled" or "remembered" ensemble. This is not about the perceptual representation and the data do not speak to that.

Reviewers' comments:

Reviewer #1 (Remarks to the Author):

Oh et al. tested the hypothesis that the PFC and PPC are using different encoding schemes in visual working memory task. This was done by applying the decoding algorithm on the broadband scalp EEG signal from the electrodes divided in the anterior and posterior set. While the question of different encoding schemes applied in different modules has relevance for mechanistic understanding of visual working memory, multiple problems with methodology and presentation make the interpretation of results questionable.

Response: We appreciate your time and effort for reviewing our manuscript. We believe that most of your concerns originated from the lack of clarity. In the revised manuscript, we have done our best to clarify the methods and results, as well as to refine the theoretical implications. Note that we highlighted major changes with different colors (blue for theoretical issues and red for methods/procedures/rationales/results).

Major comments:

1) The key parameters of assessing the decoder are missing in this paper, such as sensitivity, specificity and ROC and its AUC. The authors should show the summary plot of tuning curves, which should reveal if there is a spatial clustering of electrodes highly contributing to decoding. In addition, the assumption that the signal on posterior electrodes could be specifically ascribed to PPC, and not to a variety of visual system areas is questionable. The absence of more precise source localization makes the anatomical claims less valid.

Response: Two concerns are raised in this comment, and they are discussed separately below.

First, in the revised manuscript, we provided more information so that readers can assess the decoder. This information includes how the decoder was built, how the sensitivity was obtained, and so on. In brief, we used the inverted encoding model (IEM) to create the orientation decoder. The use of IEM in conjunction with EEG has been widely used to characterize working memory representations. Because IEM is different from other decoding methods based on binary classification, we were unable to provide its ROC and its AUC. Instead, the encoding models assume that EEGs measured from scalp electrodes reflect the neural activities of a set of *channels*, and each channel is tuned for a different orientation. Once we estimate the relative contribution of the orientation channel to each electrode on the scalp, we can build a decoder that results in responses of *all orientation channels* from EEG signals. This feature of IEM allows us to evaluate selectivity and to calculate selectivity based on the slope of the tuning function. In the revised manuscript, we added a figure explaining IEM to provide more information on the method. You can find the relevant information in Fig 3 and the section, *Reconstructing mnemonic representations of single and mean orientations* (lines 164-221)

Second, thank you for pointing out the problem, to which we were blind due to focusing too much on the PFC/PPC distinction. We cannot make a strong claim about the EEG sources based on the spatial configuration of scalp electrodes due to poor spatial resolution of EEGs. In particular, we do agree that we are unable to distinguish neural activities of PPC from those of visual areas in our study, which we have made clear in the revised manuscript. In addition, source localization was not a reasonable option for the study because we only have 28 channels and no anatomical images from individual participants. Also our source localization algorithm suitable for multivariate analyses is in its infancy (Gohel, Lim, Kim, Kwon, & Kim, 2018). However, to benefit the macroscopic information obtained from human EEG, we decided to investigate the structured representations in the brain by separating electrodes, since this is the best option for relating to neuroimaging and single-unit studies with human EEG.

To address this issue, we revised the manuscript as follows. First, we discussed the theoretical debates on PPC, and another debate on distinguishing PPC and early visual areas. We interpreted the results taking both contexts into consideration, because we cannot distinguish the PPC from visual areas in posterior sets of electrodes. You can find the relevant information, especially at the last paragraph of the discussion section (lines 461-474). Second, we performed a searchlight-like analysis to identify the critical electrodes from a subset, and to narrow down the spatial selectivity. Nevertheless, we could not find any distinguishable differences between electrodes close to parietal regions and those close to visual areas, which is described in the Supplementary materials, *Supplementary discussion 2: Topographical representations of stable and dynamic coding*.

2) The results failed to challenge the view that PFC and PPC distinction is based on the level of abstraction (Line 303). The basic sensory information (orientation) and its ensemble representation (i.e. orientation mean) could be encoded in both the homogeneous and heterogeneous during the retention period, after the subjects were

trained well to retain the mean orientation in order to get correct feedback afterwards. As a result, it is hard to differentiate whether the information encoded in the anterior region pertains to orientation or ensemble representation or is it the epiphenomena of training.

Response: We believe this concern was raised because of lack of clarity in our manuscript. In the revised manuscript, we have elaborated on previous theoretical issues about the level of abstraction. In brief, we concluded that the PFC holds task-relevant, abstract information. You can find relevant theoretical discussion in lines 412-424

We emphasize that we did NOT train participants to report mean orientations. This experiment is different from previous neurophysiological studies where categorical representation is formed through extensive training from sensory representations. This experiment is also different from previous studies asking the mean orientation from a set of visual stimuli. The feedback given in Experiment 2 was only supposed to encourage participants to remember the cued, *veridical* orientation, *not the mean* orientation. Furthermore, mounting evidence indicates that we formed ensemble representation effortlessly and feature similarity is a key factor in forming that ensemble (Son, Oh, Kang, & Chong, 2019). We emphasize this aspect in several places of the manuscript as well, especially in lines 412-424.

3) We are not getting any sense on what the stable coding looks like in the EEG signal, except that it contains the information sufficient for decoding at better than chance level. Is it a persistent oscillation in certain frequency range, consistent phase offset between the signals on different electrodes, some other signal feature(s) or feature combination?

Response: The lack of clarity in our manuscript is partly responsible for this confusion. In brief, the stable and dynamic coding of the manuscript is only defined in the context of temporal generalization, which is *indifferent* to any particular feature of EEG signals. In the revised manuscript, we clarified these terms and temporal generalization more clearly while describing the results (see the section, *Temporal dynamics of mnemonic representations of single and mean orientations*, lines 225-259).

Nevertheless, while it is impractical to explore all possible frequency and phase combinations to investigate what feature(s) of EEG signals carried the representations, it is an important question to ask which frequency band is most informative. Previous studies have shown that different frequency bands carry different types of representations (e.g. alpha-band power for spatial location). We therefore conducted a similar analysis with different frequency bands of EEG signals and we found that theta- and alpha-band spectral power appears to carry the orientation selective representation at the encoding and retention interval, respectively. However, they did not provide any consistent pattern across different experiments and conditions. This additional analysis is included in the Supplementary materials, *Supplementary discussion 5: Time-frequency representations*.

4) All of the key terms, including the dynamic and stable population coding, should be unambiguously defined at the outset. Considering the breadth of Nature Communications readership, the terms such as 'ensemble representation' - in this context used to describe the neural coding of higher order statistics from the set of visual stimuli - could be understood by many readers as denoting the stimulus representation at neural ensemble level. In addition, usage of different labels for the same concepts makes the paper difficult to follow. For example, 'homogenous' and 'inhomogeneous' trials are in some instances labeled as 'orientation' and 'ensemble'.

Response: Thank you for pointing out this weakness of our manuscript. In the revised manuscript, we clarified the terms from the onset. In addition, we also found that "homogeneous" and "heterogeneous" were not effective and, thus, we used SO (same orientation) and VO (varied orientation) throughout the manuscript to indicate the stimulus condition.

5) If the hypothesis pertains to differential encoding schemes in PFC and PPC, why was the decoding on Fig. 2 and 3 done on the electrode set that included both anterior and posterior electrodes?

Response: We first applied the IEM over all electrodes because we needed to ensure that it is suitable for studying the ensemble mean being held in working memory before investigating the structured representations of visual working memory over different brain areas. We believe this set of analyses make the method/procedure and the results more convincing.

6) It seems problematic to assume that the short autocorrelation time constant of broadband EEG (referred to as 'stable coding') implies that the 'same neural population represents the target for that period' (Lines 74-76). For example, different subsets of hippocampal place cells could encode different parts of the environment, while the theta rhythm in the local EEG would be stable.

Response: We appreciate your comments. In the revised manuscript, we carefully elaborated the definition of stable coding as well as dynamic coding.

Minor comments:

- 1) *Figure captions should start with a headline describing the results, not just describing which variables are shown on the figures.*
- 2) *The authors claim to be using the 10-20 electrode placement system, which doesn't seem to be the case, based on the electrode labels.*
- 3) *Fig. 1. A and C. It would help to put the arrow that denotes the sequential order of different trial epochs. Also, the meaning of orange and purple color codes is not denoted in the figure caption.*

Response: Thanks for pointing out these details. We have addressed all of these concerns in the revised manuscript.

- Gohel, B., Lim, S., Kim, M. Y., Kwon, H., & Kim, K. (2018). Dynamic pattern decoding of source-reconstructed MEG or EEG data: Perspective of multivariate pattern analysis and signal leakage. *Computers in Biology and Medicine*, 93, 106–116. <https://doi.org/10.1016/j.combiomed.2017.12.020>
- Son, G., Oh, B.-I., Kang, M.-S., & Chong, S. C. (2019). Similarity-Based Clusters Are Representational Units of Visual Working Memory. *Journal of Experimental Psychology : Learning , Memory , and Cognition*.

Reviewer #2 (Remarks to the Author):

This paper explores whether working memory representations are neurally coded in a stable or dynamic format. The authors use an ensemble paradigm, whereby observers had to maintain an average orientation in working memory. Because ensembles are represented hierarchically, this allowed for the exploration of working memory stability at multiple stages of processing. In two experiments, observers viewed homogeneous and heterogeneous sets of oriented stimuli and had to decide whether a test stimulus was a member of the preceding set (Experiment 1) or adjust a test stimulus to match the orientation of a cued set member (Experiment 2). The authors trained a decoder on the homogeneous condition to identify the viewed orientation based on a weighted combination of EEG electrode site activation. They then used TG matrices to determine how effective the decoder was in identifying the viewed orientation at various time points when trained on one specific time point. Finally, they tested how training on the homogeneous set generalized to the heterogeneous sets for both anterior and posterior electrode sites. Overall, authors claim the decoder showed dynamic coding for homogeneous sets and relatively stable coding for heterogeneous sets. When broken down by electrode sites, anterior regions (what authors claim to reflect PFC) were more stable for both homogeneous and heterogeneous sets while posterior regions (PPC) were more dynamic (although the results for posterior were somewhat inconsistent). Authors cite these results as evidence for differential working memory representation structures across different regions of the brain.

These experiments demonstrate a sophisticated and clever approach to exploring the nature of working memory representations. I also commend the authors on their validation technique for determining whether observers were actually representing the average orientation and not just a member of the set. Admittedly, I am not in a position to evaluate whether the decoder was implemented correctly, but I can evaluate the logic of the approach along with whether the interpretation has merit. Here are some of my concerns:

Response: We appreciate your time and effort for reviewing our manuscript. We believe that most of your concerns originated from the fact that the manuscript was not written with clarity, and information was insufficiently described. In the revised manuscript, we did our best to clearly provide sufficient information about the methods, results, and their theoretical implications. Note that we highlighted major changes with different colors (blue for theoretical issues and red for methods/procedures/rationales/results).

I am somewhat confused about how the results of the TG matrices, which showed either dynamic or stable ensemble working memory representations, are to be interpreted. What does it mean to have a dynamic working memory representation if that representation is unchanging for the duration of a trial? Why would I have to recruit different neural networks to maintain an unchanging item? Are there advantages to one system being stable as opposed to dynamic? I think some of these concerns can be addressed in an elaborated introduction section, which was somewhat lacking in its breadth (and sometimes confusing, especially the last paragraph). Many of the concepts introduced are insufficiently described, and if authors want to reach a wider audience, care should be taken to more fully describe these challenging methodological approaches.

Response: Thanks for the comments. In the revised manuscript, we elaborated the methods and theoretical implications in various places. In particular, we described the temporal generalization while describing the results and discussed general theoretical implications of stable and dynamic coding in holding working memory (see the section, *Temporal dynamics of mnemonic representations of single and mean orientations*, lines 225-259)..

Note that the question, “*why would I have to recruit different neural networks to maintain an unchanging item?*” is related to functional roles of stable and dynamic coding in working memory. In particular, the presence of dynamic coding in the PFC is challenging in explaining how we can read working memory representations from ever-changing neural patterns. In the revised manuscript, we address functional roles of stable and dynamic coding in the discussion (lines 444-460).

Discussion:

Why does it follow that stable coding vs. dynamic coding for ensembles indicates task general vs. task specificity? What does it mean to have an ensemble dynamically coded, if the representation doesn't change over the time observers are maintaining it?

Response: This concern pointed out two different questions. First, we discussed the theoretical implication of our study in the context of task-general and task-specific processes that were proposed by Freedman and his colleagues based on neurophysiological studies. In the revised manuscript, we distinguished functional roles of the PFC and PPC in different contexts (abstract representations including control and action variables vs. visual representations modulated by top-down signals) despite the fact that this new contextual information is comparable to what task-general and task-specific mean

(lines 412-431). This is because they are easy to communicate and are more relevant to recent advancements in theories of PFC and PPC.

The second question is related to the concern mentioned above, and we address this issue in the Discussion of the revised manuscript.

The explanation for the 'reconfiguration' of PPC depends on the homogeneous task being more challenging than the heterogeneous task, correct? Is there behavioral evidence to support this? In almost all other research comparing homogeneous and heterogeneous performance, observers do better when the set is homogeneous (sometimes a set size effect emerges where heterogeneous gets slightly better, but these are uncommon and hard to replicate).

Response: Thank you for pointing this out. We used the term "difficulty" incorrectly. In the revised manuscript, we corrected them. In brief, the homogeneous trials are not more difficult than the heterogeneous trials. Instead, in the homogeneous trials, participants were required to encode the orientation more precisely relative to the heterogeneous trials. This is because the probe item was sampled from more similar items, and feedback was provided in the homogeneous trials but not in the heterogeneous trials. We incorrectly described the difference in demand in terms of difficulty in the earlier submission.

Note, however, that this difference between the conditions in Experiment 1 makes it difficult to directly compare the behavioral performances between them. On the other hand, in Experiment 2, we found higher precision from the homogeneous trials than the heterogeneous trials. In the revised manuscript, we included the behavioral results of the homogeneous trials of Experiment 1 and 2, so that reviewers and readers can compare the behavioral results. The relevant information and the result of homogeneous trials can be found in *Stimuli, tasks, and behavioral results* section in lines 92-162.

Beware of making causal inferences from the decoding data (page 16, line 337)

Response: Thanks for pointing this out. In the revised manuscript, we did our best not to mix what we can infer from what we can describe.

Methods:

There is an unusually large number of participant exclusions (nearly a third of your sample). This is not typical of EEG studies, so further justification for this is warranted.

Response: Since this concern is somewhat related to the next one, we address both below.

These experimental sessions were exceptionally long (3.5 hours). How did authors ensure participants were compliant and attentive throughout this time?

Response: Thank you for pointing out this issue. Many trials are required to obtain a sufficient number of homogeneous trials to build a decoder for eight different orientations, and to obtain a reasonable number of heterogeneous trials to evaluate the neural coding of ensemble representations. Furthermore, because we decided to separate the electrodes, we needed to have as many artifact free trials as possible. With this constraint, we need to optimize the procedure so that we (a) can finish the experiment fast enough whilst (b) the participants maintain a similar level of performance throughout whilst (c) collecting as many trials as possible. Note also that the majority of the artifacts are eye movements and blink related signals, and they tend to increase with time. With these constraints, we tended to throw out more participants' data based on the artifacts in the signal.

Nevertheless, we would like to mention several things that are not usually described in the methods section. First, although the entire experiment ran for approximately 3.5 hours, this includes time for preparation and participants' hair cleaning, which is now added to the method section. Actual experiments lasted approximately 2.0 hours. Second, to make the experiments less boring for that long period of time, we played music. Third, we excluded 18 participants out of 62 for both Experiment 1 and 2. Five participants did not complete the experiment due to noise (4 in Experiment 2) as well as recording system failure. We therefore could not analyze the data.

If we excluded those five participants, 13 data sets were not included in the analyses out of 57, complete data sets (approximately 22%). This number is *not unusual* for long EEG studies. For example, Fukuda and Woodman (2017) recruited 68 participants for their three experiments and excluded 12

(approximately 17.6%) in which they excluded the data set if more than 30% of trials were lost due to oculomotor artifacts. Considering that our criterion was more stringent (20%), the 22% exclusion rate makes sense.

Note also that excluding approximately 25% of the data from analysis is quite common from my own experiences, especially with the stimulus condition that can elicit eye movements. Simply, providing details of the number of participants excluded from the analysis is a recent addition required for publication. If we look at the literature published around 10 years ago, it is easy to find papers where similar information is present.

Fukuda, K., & Woodman, G. F. (2017). Visual working memory buffers information retrieved from visual long-term memory. *Proceedings of the National Academy of Sciences of the United States of America*, 114(20), 5306–5311. <https://doi.org/10.1073/pnas.1617874114>

I don't quite understand the logic of how making the probe ranges narrower in homogeneous trials makes the orientations more memorable. Please further explain this.

Response: We did not think that a narrower probe range makes the orientation more memorable. Instead, we designed Experiment 1 so that participants make more effort in the homogenous trials by narrowing the probe range. In the revised manuscript, we clarified any places leading to misunderstanding, and we have included behavioral results for both homogenous and heterogenous trials.

The circular median isn't sufficiently explained. Is this just accounting for the fact that orientation is a circular space (i.e., 1° is actually close to 179°)? Perhaps more importantly, can authors please elaborate on what the median split is designed to achieve?

Response: Thank you for pointing this out. In the revised manuscript, we have elaborated what we sought to achieve with the median split design (lines 294-310) and added a panel in the figure to illustrate the median split design itself (Fig 5a). Despite the importance of this analysis, we failed to deliver core message in the earlier submission. We believe that this additional figure will clarify the idea and motivation of the median split analysis.

There are many decision points in the implementation of the IEM. Citations and further justification for these decisions are warranted.

Response: Thank you for pointing this out. In brief, we implemented IEM itself based on previous studies while we conducted pilot studies to determine experimental protocols (e.g. number of studies, size of stimuli and so on). We clarified and cited as best as possible where there was discussion about the how decisions were made for the implementations of IEM.

Perhaps this is implicit in the IEM implementation, but how can authors be so certain of the source of their anterior/posterior electrode activations to the point that they claim their results are primarily driven by differences between the PFC and PPC?

Response: We thank you for pointing out this weakness of the present study, and would like to add that the same issue was raised by all three reviewers. In particular, while we believe that it is reasonable to assume that anterior electrodes reflect brain signals mainly from PFC, EEG signals from posterior electrodes are likely to reflect signals from both PPC and other visual areas. In the revised manuscript, we discuss these limitations and have refined the theoretical claims in relation to the anatomical sources. In addition, we conducted a searchlight-like analysis for subsets of electrodes in order to evaluate selectivity. These results are now discussed in the supplementary materials.

Nevertheless, we emphasize that macroscopic information provided by human EEG signals are valuable despite its poor spatial resolution. In neurophysiology nowadays, we can record multiple brain areas and multiple cells, but can only provide a microscopic view from local circuits. The requires another layer of the model to put different brain areas together. We emphasized this aspect at the last paragraph of Discussion (lines 461-474).

Other concerns:

The task as described in Figure 1 is insufficient. What is the old/new judgment based on? This becomes clearer in the text, but the figures and their captions should stand on their own.

Response: Thank you for pointing out this basic error. In the revised manuscript, we completely changed the figures of behavioral results and their captions so that they stand out their own.

How were the incremental steps of the test stimulus selected? Was there pilot testing?

Response: We explored the old/new judgment for ensemble studies around the time when we started the experiments for this manuscript (Oh & Kang, 2017). It is therefore fair to say that we had some knowledge of the incremental steps of the test stimulus without running a separate pilot experiment specifically for Experiment 1.

Oh, B. I., & Kang, M.-S. (2017). Time is needed for memory to be biased toward an ensemble average. *Journal of Vision*, 17(10), 350-350. (Vision Science Society Abstract)

In the continuous response experiment, the subjects were instructed to adjust the stimulus to match one of the 4 items presented, correct? This is not entirely obvious in the text, currently.

Response: In the revised manuscript, we have clarified the procedure in the text, in addition to the text in the method section. We also added a figure illustrating the stimulus condition to dispel the ambiguity. In brief, the participants were instructed to adjust the stimulus to match the single, *cued* item and the same orientation was repeated four times.

It might be helpful to include a legend in Figure 5.

Response: In the revised manuscript, we carefully examined the figures and their captions so that readers and reviewers can easily understand the information they convey.

Jason Haberman

Reviewer #3 (Remarks to the Author):

The authors present a clever experiment investigating the neural basis of ensemble representations in memory. The encoding/decoding approach is very interesting and the design is novel. There are concerns about the stimulus choice, the comparison between homogeneous and heterogeneous, and whether the results reflect a true ensemble representation. More quantitative comparisons between conditions would also help. With revision, the manuscript will be a great addition to the literature.

Response: We appreciate your time and effort for reviewing our manuscript, and your encouraging comments. We have done our best to address all of your concerns by adding an index which allows us to compare the results quantitatively, and by increasing the clarity of the manuscript. Note that we highlighted major changes with different colors (blue for theoretical issues and red for methods/procedures/rationales/results).

Fig. 1 Where's the data for the homogeneous condition? It's important to compare the homogeneous to the heterogeneous condition.

Response: We included the results from the homogenous condition in the revised manuscript (Fig 2a & 2c, lines 146-162).

How do we know observers don't just sample one gabor patch in the heterogeneous condition? If observers picked one gabor patch and based decision on that (old/new or estimation/matching/adjustment), their data would look similar to that in Fig 1. On average, across trials, sampling single Gabor patches would produce data that is similar to that in Fig. 1. In other words, what is the integration efficiency?

Response: Thank you for pointing out this issue.

Several studies have shown that we do integrate only a subset of items to build an ensemble representation, and integrating the information from a subset of items is evidence for ensemble (Whitney & Yamanashi Leib, 2018). But, sampling only one item is evidence against the ensemble (Whitney & Yamanashi Leib, 2018).

Now, we address a special case of subsampling hypothesis, sampling only one. First, let us illustrate this point with Experiment 2. It is difficult to imagine that we can only select one item despite the fact that the probability of that item being cued is only 1/20. Second, let us still imagine that one can select only one item out of 20 items. For example, if the selected item is -22.5 deg one, the estimated bias should be 22.5 deg. If the selected item is -7.5 deg one, the estimated bias should be 7.5 deg, and so on. However, the estimated biases were smaller than the prediction by half. We therefore conclude that it is very unlikely that participants would select just one item among 20 possible items. Third, most importantly, sampling a single item cannot explain the median-split analysis results. If participants selected only one item, their response should be very close to that item. Nevertheless, when we applied the decoder to those subset of trials, orientation/ensemble selective responses disappeared. Detailed rationale of the median-split analysis was elaborated on in the revised manuscript (lines 294-310).

Fig 1D. Are the light colored dots individual subjects? The figure caption is unclear. It says "individual data." What is "individual data"? Individual subjects? Sessions? Individual blocks? Individual trials (too few to be that, I assume, but it's unclear).

Response: They are data points of individual participants. We have clarified them in the revised manuscript.

Fig. 2A. Does "orientation" refer to only the homogeneous condition? The mapping of homogeneous and heterogeneous onto the "orientation" and "ensemble" is unclear in Fig 2.

Response: Thank you for pointing this out. In the revised manuscript, we have now changed the term by using SO (same orientation) and VO (varied orientation) to refer to homogenous and heterogeneous trials. In addition, we neutralized the use of SO and VO in order to refer to them as stimulus conditions rather than representations. This means that we distinguished between the stimulus conditions, names of decoders, and the representations as best as we could.

Fig. 2B. If this same analysis were done for all orientations around the mean (not just the mean), what would decoding performance look like? Is decoding best specifically for the mean, or are all orientations in the

heterogeneous display decoded just as well? If the decoding is sharply tuned to average orientation, that would be nice to know.

Response: IEM does not build a decoder for individual orientations. *We build one decoder for all eight orientations.* Therefore if two items were presented, both can be separately decoded at the same time if they are far apart in feature space or if a single, broad tuning is obtained if they are close to each other in the feature space (Sprague, Ester, & Serences, 2014; Sutterer, Foster, Adam, Vogel, & Awh, 2019).

In the heterogeneous condition of our experiment, four similar items were presented whose range was comparable to distances between three adjacent channels. Therefore, the reconstructed channel response from the heterogeneous stimuli should be the sum of four orientations whose center corresponds to the ensemble mean. Theoretically, the ensemble should produce sharper tuning than four orientations. Nevertheless, because we cannot determine the tuning of the ensemble mean or four similar orientations, we performed the median split analysis to determine whether the orientation-selective response was the ensemble mean or four orientations. Our conclusion was that the reconstructed orientation should be closer to ensemble mean.

In the revised manuscript, we described the IEM in detail (lines 164-206), and clarified the purpose of the median-split analysis (lines 284-310).

Fig. 4. Why separate electrodes in this way? Seems arbitrary. The comparison of temporal tuning for the different arrays of electrodes also seems descriptive rather than quantitative. Many of the results are presented as visual comparisons of heatmaps, which is not ideal.

Response: Since this is related to the next concern, we address them both below.

Line 236-238. The statements here don't seem well supported by the data, or at least are vague descriptions without any direct statistical comparison. E.g., "...spread out broadly across the horizontal axis..." Everything is spread out broadly across the horizontal axis... We need a more precise metric of the tuning and a direct comparison between panels. We also need a qualification that this is based on a very arbitrary division of electrodes in anterior and posterior.

Response: Thank you for pointing this out. To address this concern and the one above, we introduced an index and conducted a searchlight-like analysis for a subset of trials. First, we calculated two types of index for the stable and dynamic coding formats that summarized the two-dimensional TG matrix into a single-dimensional, the magnitude of the coding format over time. Second, to study different coding formats of working memory over distributed over brain areas, we separated the electrodes into two halves. We have shown that selection of different electrodes did not change the pattern of results in the original submission as long as those two sets of electrodes covered broadly anterior and posterior parts of the scalp electrodes. Here, we advanced one step further such that we selected a subset of neighboring electrodes (i.e. Fz, FC1, FC2, C3, C4, CP1, CP2 & Pz are selected as the neighboring electrodes for Cz) for all electrodes and conducted the same analyses to examine the spatial selectivity of the stable and dynamic coding. We found that the stable coding was still dominant in the anterior electrodes while the dynamic coding was dominant in the posterior electrodes with different time courses. The results of these new analyses are included in the Supplementary materials.

Was there source localization? How do we know the electrodes genuinely reflect anterior and posterior cortical activity?

Response: Thank you for pointing out this limitation. The spatial resolution of EEG signals is poor. We only had 28 channels for our scalp electrodes. While there are several procedures for estimating sources which have been proposed based on the activation-based signals, it is still difficult to determine sources from small numbers of channels. Furthermore, creating the source space for information-based analyses such as multivariate analysis is in infancy (Gohel et al., 2018).

Nevertheless, whilst we are unable to precisely localize the neural sources of working memory representations to the degree of fMRI or single unit recordings, our procedure is a reasonable choice to investigate structured working memory representations over distributed brain regions. In particular, we emphasize that macroscopic information provided by human EEG signals contain valuable information which cannot be obtained even from multi-unit recordings over multiple brain areas. This is because such recordings provide a microscopic view from local circuit which requires another layer of the model to put different brain areas together.

Note, however, that the selected posterior electrodes cannot distinguish neural sources between PPC and other visual areas. The searchlight-like analysis provided the same picture. We were blind to this aspect in the initial submission, and this was pointed out by reviewer #1. In the revised manuscript, we have discussed the limitations of our study (lines 461-474).

The cue is unclear in methods and in main text. Was the cue always in the same spatial location? There was always only one cue?

If the cue was in a predictable location, it seems that subjects could attend to that location, and probably simply maintain and report that single orientation. It would be very surprising to find bias in reported orientation for an oriented Gabor patch in a known location, unless there was crowding. But perhaps the cue location is randomized in some way.

Assuming the cue was in an unpredictable location, then the question is whether subjects actually integrated more than one gabor patch or just relied on one oriented stimulus to make their judgments. Across trials it would look like an ensemble, once data is averaged, but on any single trial it might be a single oriented Gabor patch that drives responses. The data in Fig 1D do not address this. Those colored dots represent averaged data, so on any trial subjects may have used just one stimulus. There may be a way to simulate performance or analytically prove that observers used multiple stimuli in their ensemble estimates. This is foundational for all of the results

Response: In the revised manuscript, we have clarified the experimental procedures. In brief, the cue location of Experiment 2 was unpredictable from trial to trial. The central question of this concern is whether participants did form ensemble representations. Above, we ruled out a subsampling hypothesis.

The brain could use ensemble representations for both heterogeneous and homogeneous sets of stimuli. The difference is really the variance of the display, not necessarily whether an ensemble is used. The IEM is very clever, and it may make sense to use homogeneous (or perhaps better to use single stimuli) to build that decoder. However, when the data is presented in the figures and discussed, it is presented by comparing "orientation" versus "ensemble" panels. Yet, homogeneous and heterogeneous sets of stimuli may be represented as ensembles, so the orientation data (e.g., Fig 2A,C, E,G) may reflect ensemble representations. Any difference is really about the variance in the display. In fact, homogeneous displays may be better represented/encoded/maintained as an ensemble. Increasing variance tends to reduce sensitivity to ensemble characteristics in many domains. One useful approach would be to investigate how much orientation variance needs to be added to the display to change the data in Fig. 1E toward Fig. 1F.

Response: We do agree with the statement: "In fact, homogeneous displays may be better represented/encoded/maintained as an ensemble." This is therefore why we concluded that the PFC holds abstract representations even though it is difficult to define the abstract representation of a single orientation from the perspective of the stimulus.

We therefore introduced the variability in orientation and focused on ensemble mean to utilize the power of IEM for studying the mean of ensemble representations. However, note that growing evidence suggests that our brain represents the mean as well as the variance of ensemble representations (Michael, de Gardelle, & Summerfield, 2014) and even any characteristics of distribution (Chetverikov, Campana, & Kristjánsson, 2016).

Nevertheless, we believe that the suggested experiment is beyond the scope of the present study. If we understand the experiment correctly, the suggestion implies an experiment that creates a neurometric curve from dynamic coding to stable coding while systematically varying the variance of the stimuli. This experiment is appealing but impractical when considering that participants spent approximately 3.5 hours just to complete two conditions. In addition, we have shown that stable coding was dominant for both homogenous and heterogenous trials across the two experiments, and the coding formats of the occipitoparietal electrodes depend on the task demand. This means that the theoretical benefit for varying the variances of the stimuli is ambiguous.

"...The dominant contribution of anterior brain regions to the ensemble representations." Specify that this about the "maintained" or "recalled" or "remembered" ensemble. This is not about the perceptual representation and the data do not speak to that.

Response: Thank you for raising this issue. In the revised manuscript, we carefully distinguished between any ambiguities.

- Chetverikov, A., Campana, G., & Kristjánsson, Á. (2016). Building ensemble representations: How the shape of preceding distractor distributions affects visual search. *Cognition*, 153(C), 196–210. Retrieved from <http://dx.doi.org/10.1016/j.cognition.2016.04.018>
- Gohel, B., Lim, S., Kim, M. Y., Kwon, H., & Kim, K. (2018). Dynamic pattern decoding of source-reconstructed MEG or EEG data: Perspective of multivariate pattern analysis and signal leakage. *Computers in Biology and Medicine*, 93, 106–116. <https://doi.org/10.1016/j.combiomed.2017.12.020>
- Michael, E., de Gardelle, V., & Summerfield, C. (2014). Priming by the variability of visual information. *Proceedings Of The National Academy Of Sciences Of The United States Of America*, 111(21), 7873–7878. Retrieved from <http://www.pnas.org/cgi/doi/10.1073/pnas.1308674111>
- Sprague, T. C., Ester, E. F., & Serences, J. T. (2014). Reconstructions of Information in Visual Spatial Working Memory Degrade with Memory Load. *Current Biology : CB*, 24(18), 2174–2180. Retrieved from <http://dx.doi.org/10.1016/j.cub.2014.07.066>
- Sutterer, D. W., Foster, J. J., Adam, K. C. S., Vogel, E. K., & Awh, E. (2019). Item-specific delay activity demonstrates concurrent storage of multiple items in working memory. *PLoS Biology*, 17(4): e30. <https://doi.org/10.1101/382879>
- Whitney, D., & Yamanashi Leib, A. (2018). Ensemble Perception. *Ssrn*. <https://doi.org/10.1146/annurev-psych-010416-044232>

Reviewers' Comments:

Reviewer #1:

Remarks to the Author:

The revised manuscript helped clarify some of the misunderstandings, such as the nature of stable and dynamic coding, as well as the structure of visual stimuli. The authors acknowledged the fact that the EEG signal source localization with the set up used could not reliably ascribe the posterior activity to PPC, rather than to adjacent visual areas. They also made the effort in clarifying the underlying electrophysiological basis of visual memory encoding, although without a very clear result. This is unfortunate, as it seems that the only advantage taken from an information-rich EEG signal is its temporal resolution.

Some of the critical points are still not adequately covered in the revised version.

1. We understand that the performance of the IEM decoder cannot be assessed by AUC or ROC. However, it is crucial to demonstrate the decoder reliability, as it forms the basis for multiple analyses, and some of the performance metrics (other than AUC or ROC) could be applied, such as the goodness-of-fit [1]. It is possible that the less sensitive 'target' response (Lines 314-323) could result from the low decoder performance.
2. Authors need to discuss the assumptions of the IEM method and possible alternative interpretations of these results.

[1] Liu, Taosheng, Dylan Cable, and Justin L. Gardner. "Inverted encoding models of human population response conflate noise and neural tuning width." *Journal of Neuroscience* 38.2 (2018): 398-408.

Reviewer #2:

Remarks to the Author:

The authors have done a commendable job attempting to address all of my concerns raised in the first submission. Overall, this is a novel approach addressing an interesting research question related to ensembles and working memory. I have just a couple, global concerns that I hope authors are willing to address in a future revision.

- I am still not clear on the implications of stable or dynamic coding, as raised in my original review. Authors do a nice job conceptually describing the difference between a temporally stable representation versus a dynamic one, however, I do not know what that means functionally. If I am maintaining a single representation in working memory, what are the behavioral consequences of having it stably represented versus dynamically? Authors do mention in the discussion something about creating a more honed representation moving from PPC to PFC, but I (and, I imagine, your readers) would appreciate a more detailed discussion of what research has been done exploring what these representational differences mean from a functional perspective.

- Did authors expect these exact results? This is very much exploratory in nature, which I think is valuable scientific practice, but one must use caution in post-hoc speculative interpretations. How confident are the authors that, if they ran the exact same study again, these results would replicate as described?

- I appreciated the manipulation designed to distinguish coding of individual items vs. the ensemble mean. This is sometimes hard to distinguish, and so I was pleasantly surprised that the decoder worked so effectively.

Minor points:

- Authors might consider adding a legend for colors of median split analysis in Figure 5a.

- Discussion: authors mention that their work is different from previous studies because they emphasized remembering the mean using a method-of-adjustment, but there are, in fact, many studies that employ the method-of-adjustment asking participants to adjust to the mean only (e.g., Haberman & Whitney, 2010; Haberman, Brady, & Alvarez, 2015).

Jason Haberman

Reviewer #3:

Remarks to the Author:

Thanks for addressing all of my concerns.

Reviewers 1's comments:

Reviewer #1 (Remarks to the Author):

The revised manuscript helped clarify some of the misunderstandings, such as the nature of stable and dynamic coding, as well as the structure of visual stimuli. The authors acknowledged the fact that the EEG signal source localization with the set up used could not reliably ascribe the posterior activity to PPC, rather than to adjacent visual areas. They also made the effort in clarifying the underlying electrophysiological basis of visual memory encoding, although without a very clear result. This is unfortunate, as it seems that the only advantage taken from an information-rich EEG signal is its temporal resolution.

Response: We appreciate your time and effort in reviewing our manuscript. We are glad that our revision clarified the manuscript.

While our efforts to narrow down the spatial selectivity failed to distinguish working memory representations in PPC and visual areas, our goal is to maximize the advantage of EEG signal rather than being discouraged by its poor spatial resolution. As we are trained as experimental psychologists, we believe that we can address the roles of those two areas in conjunction with novel behavioral protocols and we recently made some advancement, which will be presented in *Psychonomics* this year. We hope that we can contribute to the literature in a near future.

Below, we addressed your remaining concerns.

Some of the critical points are still not adequately covered in the revised version.

1. We understand that the performance of the IEM decoder cannot be assessed by AUC or ROC. However, it is crucial to demonstrate the decoder reliability, as it forms the basis for multiple analyses, and some of the performance metrics (other than AUC or ROC) could be applied, such as the goodness-of-fit [1]. It is possible that the less sensitive 'target' response (Lines 314-323) could result from the low decoder performance.

Response: Thank you for your comments. This concern addresses two separate topics, and they are discussed separately below.

First, we concluded that goodness-of-fit is not a suitable measure of reliability of IEM when IEM is used to generalize the model of one condition to a different condition, instead of being applied to different trials of the same condition.

We calculated the goodness-of-fit based on the suggested literature. Goodness-of-fit provides the amount of variability of EEG signals explained by the model from different trials of the same condition (e.g. SO). Figure AR2 1a and 1b show the goodness-of-fit for Experiments 1 and 2, respectively. The goodness-of-fit was higher at earlier phases of the retention interval (~ 0.8) and then gradually reduced (~ 0.2). We would like to discuss two aspects.

- 1) While it is difficult to compare the values from our study to those of Liu et al. (2018), on average, our values were higher than theirs (0.31 and 0.13 for high- and low-contrast conditions).
- 2) The gradual decline in the goodness-of-fit shown in Figure AR2 1a and 1b does not necessarily mean a decline in reliability. Instead, the earlier higher goodness-of-fit can reflect the similarity in patterns of electrophysiological responses that can be driven by similar patterns of neural responses accompanied with similar states of neural responses (i.e. the same preparatory state for the eight different orientations). It means that it is difficult to compare the goodness-of-fits obtained from different times because they can reflect different cognitive processes.

Figure AR1. Goodness of fit obtained from SO condition of Exp 1 and 2. Shaded areas represent ± 1 standard error.

However, SNR can influence the decoding performance such that sensitivity becomes higher with high SNR and vice versa. This possibility can lead to a concern that the results based on sensitivity can be just an outcome of SNR instead of actual representations.

- 1) Note that the goodness-of-fit should not be applied to the SO-VO prediction. The goodness-of-fit is designed to evaluate the reliability of the model, not the performance of the model when it is applied to a different type of data set. It means that the rationale behind goodness-of-fit does not apply to the SO-VO prediction where the decoder built from the SO condition is applied to the VO condition. While both SO and VO conditions should have a common aspect (ensemble mean), their variance should be wider in SO-VO prediction than the SO-SO prediction. It means that goodness-of-fit applied to the SO-VO cannot reflect reliability. If we blindly apply the procedure, the predicted EEG signals of the VO condition by the SO decoder should be more different from the raw EEG signals of VO condition, and $\text{SUM}(\text{B}_{\text{prediction}} - \text{B}_2)^2$, the equation [5] of Liu et al. (2018), should increase. As a result, the goodness-of-fit can be reduced without any bound. It means that even a negative goodness-of-fit can be obtained and it would not reflect the reliability or SNR of the model.
- 2) One may have a concern that the correlation between sensitivity and behavioral performance shown in Figure 7 could have been the outcome of SNR rather than actual representations. Although the goodness-of-fit does not provide an index for SNR in the SO-VO prediction, we calculated correlation between the goodness-of-fit obtained from SO-SO prediction as a proxy for the SNR level of the decoder and behavior as in Fig 7a and 7b by replacing the sensitivity with the goodness-of-fit, and found that correlation was insignificant ($R^2=.005$, $p=.778$ for Exp 1 and $R^2=.041$, $p=.342$ for Exp 2)

IEM is a recent addition to the literature in multivariate pattern analyses. We believe that the community will identify a measure of reliability that is suitable when applying a model to a different set of data. Until then, we think that the sensitivity index we used based on the slope of the tuning curve can provide something similar to ROC in the classification. Specifically, ROC can be built based on hits and false alarms. In the IEM where 8 orientations can be decoded at the same time, a hit can result in a peak modulation at the target orientation and a false alarm can result in peak modulations at other orientations. If those two cases occur in a similar proportion, the slope on average should be close to zero as the ROC curve lies close to the diagonal line.

Second, we do not think that the low decoder performance resulted in the lack of “target” response for the following two reasons.

- 1) We applied the same decoder built from the SO condition to the “mean” and “target” response trials with an aim to distinguish whether the outcome of decoding represented the ensemble mean or a set of visual stimuli.
- 2) The null result is subject to a concern that the decoder was not sensitive enough to uncover underlying representations; however, the null result of the “target” response trials does not undermine our theoretical interpretation either. If IEM decoded the sensory stimuli, we should see comparable decoding outcomes between the “mean” and “target” response trials regardless of the variability in response errors. We do not rule out the possibility that participants could have held the “target” representation in a weaker strength in relative to the “mean” representation. However, this does not change the interpretation that the IEM mainly decoded the ensemble representations. Furthermore, it is unlikely that participants could have held the target orientation throughout the experiments based on previous studies on cognitive psychology and ensemble representations, which we argued in the previous revision.

3) We revised the manuscript to address this concern (lines 322-329):

“The lack of “target” responses is subject to a concern that the decoder was not sensitive enough to uncover underlying representations such that participants could have held the “target” representation in a weaker strength relative to the “mean” representation. However, this does not change the interpretation that the IEM mainly decoded the ensemble representations. Furthermore, it is unlikely that participants could have held the cued, target orientation throughout the experiments despite the fact that only 1 out of 20 locations, or 1 of 4 orientations were randomly selected for the report.”

2. Authors need to discuss the assumptions of the IEM method and possible alternative interpretations of these results.

Response: The methodological limitations of IEM have been a topic of much debate in recent years (Liu et al., 2018; Sprague et al., 2018). Specifically, IEM recovers the modeled channel responses rather than the stimulus (Gardner & Liu, 2019). It means that the shape of channel response does not reflect the shape of neural responses of the stimulus but reflects the shape of a base model. In addition, the tuning width of IEM is uninformative in relation to actual tuning of individual neurons or a set of population responses. Instead, a simulation study has shown that the tuning width is also sensitive to noise (Liu, Cable, & Gardner, 2018). However, our interpretation of the results remains reasonable for the following reasons.

First, we used the permutation approach to avoid the possibility that the choice of the base function for IEM introduced any systematic bias because we used the same base function for generating the “null distribution.” Second, we built the decoder from the SO condition and applied it to the VO condition to create the SO-VO prediction, in which the recovered channel response is not related to the actual stimuli and, thus, we can infer the ensemble representation. Third, we demonstrated that the SO-VO prediction decoded the ensemble mean based on the correlation across observers (Exp. 1-2) and a median-split analysis (Exp. 2). Fourth, the slope (the metric of selectivity) reflected the amplitude and the width of the tuning results instead of the width, which is sensitive to noise rather than actual tuning width of the representation. Lastly, the fact that the reconstructed channel responses can reflect the relative likelihood of each channel makes IEM still useful in inferring relative channel responses rather than channel tuning *per se*.

In the revised manuscript (lines 421-438), we discussed the methodological limitations of IEM and addressed the concern of IEM in relation to our study.

“Although IEM is suitable for assessing population-level mental representations⁴³, IEM is methodologically limited in that it reconstructs any arbitrary, modeled channel responses⁴⁴ and the noise and population-level of neural tuning conflate the channel responses⁴⁵. As a result, our ability to infer tuning properties (e.g. tuning width) of the neural representations from the reconstructed channel responses is limited. Nevertheless, our results remain tenable for the following reasons. We used the permutation approach to avoid the possibility that the choice of the basis function for IEM introduced any systematic bias because we used the same basis function for generating the “null distribution.” We also used the slope of the tuning function as an index of the channel sensitivity. Because the slope is an outcome of both the amplitude and width of the channel responses, our measure of sensitivity should be less sensitive to noise as well as hypothesized channel responses in contrast to the noise sensitive tuning width⁴⁵. The fact that the reconstructed channel responses can reflect the relative likelihood of each channel makes IEM still useful in inferring relative channel responses rather than channel tuning *per se*. Most importantly, we built the decoder from the SO condition and applied it to different trial types of the VO condition (“mean” and “target” response trials shown in Fig 5) to establish the ensemble representation. Together, our results remain tenable, despite the methodological limitations of IEM.”

[1] Liu, Taosheng, Dylan Cable, and Justin L. Gardner. "Inverted encoding models of human population response conflate noise and neural tuning width." *Journal of Neuroscience* 38.2 (2018): 398-408.

Reviewer #2 (Remarks to the Author):

The authors have done a commendable job attempting to address all of my concerns raised in the first submission. Overall, this is a novel approach addressing an interesting research question related to ensembles and working memory. I have just a couple, global concerns that I hope authors are willing to address in a future revision.

Response: We appreciate your time and effort in reviewing our manuscript. We are glad that our revision addressed your earlier concern. Below, we addressed your remaining concerns and shared our experiences in the revised manuscript.

- I am still not clear on the implications of stable or dynamic coding, as raised in my original review. Authors do a nice job conceptually describing the difference between a temporally stable representation versus a dynamic one, however, I do not know what that means functionally. If I am maintaining a single representation in working memory, what are the behavioral consequences of having it stably represented versus dynamically? Authors do mention in the discussion something about creating a more honed representation moving from PPC to PFC, but I (and, I imagine, your readers) would appreciate a more detailed discussion of what research has been done exploring what these representational differences mean from a functional perspective.

Response: Thank you for raising this concern.

With the advancement of decoding techniques with EEG/MEG data, stable and dynamic coding formats have been identified from multiple studies. Stable coding appears to reflect the neural representations of behaviorally relevant information possibly guided by sustained attention. On the other hand, it is true that multiple information was simultaneously found from the dynamic coding from the occipitoparietal areas, but it is premature to discuss their functional roles with confidence. In the revised manuscript, we elaborated on the functional roles of stable and dynamic coding as follows (lines 471-491)

“Another advancement of our study is the distinguishable coding formats that we identified from TG, which would contribute to the literature in shaping the functional roles of different coding formats. Studying what information is held in different coding formats identified from EEG/MEG signals is an active research area. We found stable coding from frontocentral areas in all conditions and experiments, and this stable coding also correlated with a behavioral measure of ensemble tendency. We also found stable coding at the occipitoparietal areas in Experiment 2, where attention could have been important for reporting the cued orientation.

These results are consistent with those of recent studies, suggesting that stable coding is important in guiding our behaviors in relation to the sustained attention. Target stimulus in rapid serial visual presentation elicited category-specific stable coding only when it was subsequently reported⁵². When participants needed to determine a briefly presented target (face or house) sandwiched by masks, the participants' confidence modulated the stable coding both at the frontocentral and occipitoparietal areas⁵³. In working memory literature, stable coding was modulated by selection rules at ventrolateral prefrontal regions⁵⁴ and by attentional priority at the occipitoparietal electrodes⁵⁵. Together, these results suggest that stable coding holds task-relevant information that guides our behavior through attention or metacognition. On the other hand, functional roles of dynamic coding remain elusive in the literature. However, considering that dynamic coding at the occipitoparietal areas represents both targets as well as distractors⁵² and multiple features prior to the selection⁵⁴, it reflects a “perceptual buffer”^{52,56}.”

- Did authors expect these exact results? This is very much exploratory in nature, which I think is valuable scientific practice, but one must use caution in post-hoc speculative interpretations. How confident are the authors that, if they ran the exact same study again, these results would replicate as described?

Response: Because this concern is related to the next one, we will address both below.

- I appreciated the manipulation designed to distinguish coding of individual items vs. the ensemble mean. This is sometimes hard to distinguish, and so I was pleasantly surprised that the decoder worked so effectively.

Response: Thank you for your comments.

We will first respond to the two comments here by sharing how this project has evolved over time.

We have sought to identify neural representations of ensemble because we can ask many important and interesting questions on ensemble representations in perception as well as in memory even though we did not have any specific questions in mind at first. To start, the project had an exploratory aspect.

Over the evolution of the project, we became confident that we could decode some representations from the VO condition that are different from the SO condition. We then evaluated whether the decoded representations were ensemble rather than sensory representations of individual items. The median-split analysis shown in Fig 5 played a pivotal role that gave us confidence that we can study ensemble by using IEM.

In the manuscript, we tried to share this evolution of the project such that we establish the behavioral protocol of ensemble (Fig 2), introduce and establish the SO-VO prediction based on IEM (Fig 3), and then showed the median-split analysis (Fig 5).

Let us now turn to your questions.

First, we expected the dynamic coding formats from occipitoparietal electrodes from the SO-SO prediction as a replication of many previous studies (e.g. Myers et al., 2015). In other words, this result was used as a manipulation check for this experiment. However, we did not expect other results including different coding formats between the frontocentral and occipitoparietal electrodes, correlation difference as shown in Fig 7, or different SO-VO prediction results obtained from the occipitoparietal electrodes between Exp 1 and Exp 2.

Second, although the study was rather exploratory in nature, we tried to establish key aspects of the procedure one by one. We are therefore confident that the results will be replicated if the participants performed a working memory task with similar requirements based on the two reported experiments, and some experiments we conducted at the pilot stages. Nevertheless, we are not sure whether an average task, which requires participants to report a mean orientation, would provide a similar result. We plan to compare those two different tasks in the future.

Third, we were also surprised by the outcome of the median-split analysis, which was the procedure that allowed us to distinguish the ensemble representations from the set of individual items. If we did not obtain this result, the entire project would have been stalled.

Minor points:

- *Authors might consider adding a legend for colors of median split analysis in Figure 5a.*

Response: Thank you for the suggestion. We added a legend for the “mean” response trials and the “target” response trials.

- *Discussion: authors mention that their work is different from previous studies because they emphasized remembering the mean using a method-of-adjustment, but there are, in fact, many studies that employ the method-of-adjustment asking participants to adjust to the mean only (e.g., Haberman & Whitney, 2010; Haberman, Brady, & Alvarez, 2015).*

Response: Thank you for the comment. We realized that the manuscript was not clear enough if the reviewer is referring to our discussion starting from the line 435.

We did not emphasize the importance of the adjustment procedure. Instead, we emphasized that we did NOT ask participants to report only the mean of a set of stimuli, so called average tasks. We asked participants to remember all stimuli so that we could probe working memory rather than the mean itself. To reduce confusion, we changed the relevant sentences in the manuscript as follows.

explicitly required participants to adjust to or determine the mean of a set of stimuli, where only the representation of ensemble means should eventually be held in memory²⁴

explicitly required participants to compute the mean of a set of stimuli²⁴

Jason Haberman

Reviewer #3 (Remarks to the Author):

Thanks for addressing all of my concerns.

Response: We appreciate your time and effort in reviewing our manuscript. We are glad that our revision addressed all your concerns.

Reviewers' Comments:

Reviewer #1:

Remarks to the Author:

The authors have addressed my concerns. I have no other comments.

Reviewer #2:

Remarks to the Author:

The authors have adequately addressed all the concerns I previously raised. Thank you!

Jason Haberman